Manuscript prepared for Geosci. Model Dev.
with version 4.2 of the LaTeX class copernicus.cls.
Date: 21 November 2018

# Nemo-Nordic 1.0: A NEMO Based Ocean Model for Baltic & North Seas, Research and Operational Applications

**Robinson Hordoir**[1,2], **Lars Axell**[3], **Anders Höglund**[3], **Christian Dieterich**[3], **Filippa Fransner**[2,4,5], **Matthias Gröger**[3], **Ye Liu**[3], **Per Pemberton**[3], **Semjon Schimanke**[3], **Helen Andersson**[3], **Patrik Ljungemyr**[3], **Petter Nygren**[3], **Saeed Falahat**[3], **Adam Nord**[3], **Anette Jönsson**[3], **Iréne Lake**[3], **Kristofer Döös**[5], **Magnus Hieronymus**[3], **Heiner Dietze**[6], **Ulrike Löptien**[6], **Ivan Kuznetsov**[7], **Antti Westerlund**[8], **Laura Tuomi**[8], **and Jari Haapala**[8]

[1]Institute of Marine Research, Bergen, Norway
[2]Bjerknes Centre for Climate Research, Bergen, Norway
[3]Swedish Meteorological and Hydrological Institute, Norrköping, Sweden
[4]Geophysical Institute, Bergen University, Norway
[5]Department of Meteorology, Stockholm University, Sweden
[6]GEOMAR, Helmholtz Centre for Ocean Research, Kiel, Germany
[7]Institute of Coastal Research, Helmholtz-Zentrum, Geesthacht, Germany
[8]Finnish Meteorological Institute, Helsinki, Finland

*Correspondence to:* Robinson Hordoir
(robinson.hordoir@hi.no)

**Abstract.** We present Nemo-Nordic, a Baltic & North Sea model based on the NEMO ocean engine. Surrounded by highly industrialised countries, the Baltic and North seas, and their assets associated with shipping, fishing and tourism; are vulnerable to anthropogenic pressure and climate change. Ocean models providing reliable forecasts, and enabling climatic studies, are important tools for the shipping infrastructure and to get a better understanding of effects of climate change on the marine ecosystems. Nemo-Nordic is intended to come as a tool for both short term and long term simulations, and to be used for ocean forecasting as well as process and climatic studies. Here, the scientific and technical choices within Nemo-Nordic are introduced, and the reasons behind the design of the model and its domain, and the inclusions of the two seas, are explained. The model's ability to represent barotropic and baroclinic dynamics, as well as the vertical structure of the water column, is presented. Biases are shown and discussed. The short term capabilities of the model are presented, and especially its capabilities to represent sea level on an hourly timescale with a high degree of accuracy. We also show that the model can represent longer time scales, with a focus on the Major Baltic Inflows and the variability of deep water salinity in the Baltic Sea.

## 1 Introduction

The Baltic Sea is a semi-enclosed sea that is heavily influenced by freshwater input from large continental rivers. The large freshwater input, and the narrow connection with the North Sea and the Atlantic Ocean through the Danish straits (Figure 1), gives the Baltic Sea its brackish characteristics with a strongly stratified water column, and a water residence time of ca. 40 years (Döös et al., 2004).

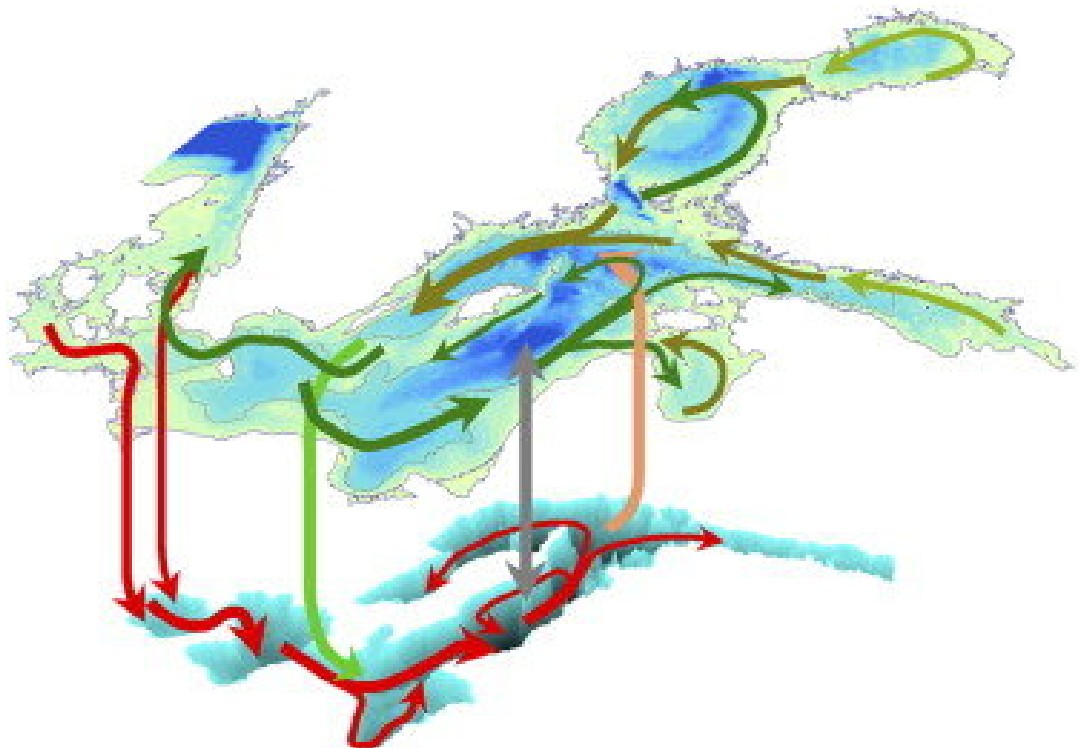

**Fig. 1.** Schematic view of the Baltic Sea circulation (From Elken and Matthäus (2008)). The Baltic Sea can be considered as a large sill estuary in which salty water masses from the North Sea may manage to flow towards the bottom. These water masses eventually mix with lighter and fresher water masses. Sub-estuarine circulations take place in the Gulf of Finland and in the Gulf of Bothnia.

Due to the long residence time and the strong stratification, the Baltic Sea ecosystem is vulnerable to anthropogenic pressure. As a result of large nutrient inputs the sea is today classified as eutrophied (Andersen et al., 2017), and a spreading of anoxic bottom waters has been observed during the last century (Carstensen et al., 2014). Eutrophication and anoxia can have severe impacts on the ecosystem. The cod, for example, which is a economically important fish species (Köster et al., 2003; Wieland et al., 1994), suffers from anoxic deep waters as its eggs reside in the Baltic Sea deep waters. Further, although all underlying mechanisms are still not clear, increases in cyanobacteria blooms are often related to eutrophication (Vahtera et al., 2007).

The vertical structure of the Baltic Sea, and its ecosystem, is on the long term influenced by inflows of salt water masses from the North Sea through the Danish Straits (e.g Boesch et al., 2006; Eilola et al., 2009; Fonselius and Valderrama, 2003; Neumann et al., 2012; Pawlak. et al., 2009; Wasmund and Uhlig, 2003; Fredrik Wulff, 1990). The time scale of a typical "Major Baltic Inflow" (MBI hereafter), which provides salt and oxygen to the deepest parts of the Baltic Sea and fuels the Baltic Sea baroclinic circulation (Döös et al., 2004), is on the order of 40 days (Schimanke et al., 2014). MBIs are barotropic in nature, and the variability of the barotropic circulation on time scales of hours to days can affect the Baltic Sea ecosystem on time scales up to several decades or a whole century.

Fueled by advances in computer hardware, the quest towards a more comprehensive understanding of the crucial ventilation of the Baltic Sea has been aided by simulations of inflows with numerical general ocean circulation models (Hordoir et al., 2015), and it has been shown that these inflows are closely correlated to the sea surface height (SSH hereafter) variability in both the Baltic and the North Sea (Gustafsson and Andersson, 2001). An ocean model that provides a consistent representation of the Baltic Sea long term ecosystem with its specific haline dynamics and stratification on one side, and that allows to make an accurate forecast of SSH for the Baltic & North Sea basin, is not incompatible: it is complementary.

The North Sea is, compared to the Baltic Sea, a dynamical region with a water residence time of only a few years (Otto et al., 1990). It is characterized by strong tidal currents and a general cyclonic large scale circulation pattern (Winther and Johannessen, 2006) with major inflow along the British Isles in the western part of northern boundary and an outflow along the Norwegian Channel in the East (Figure 2).

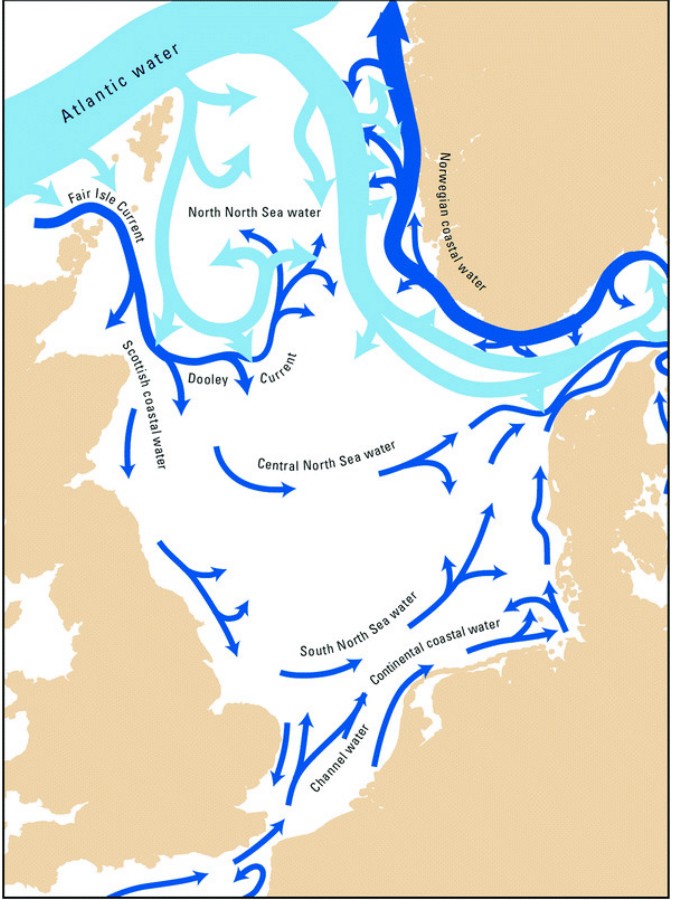

**Fig. 2.** Schematic view of the North Sea circulation (From OSPAR (2000)). The North Sea is a dominated by a cyclonic circulation (blue arrows) driven by wind and tidal forcing. The input of freshwater along South-Eastern coasts and from the Baltic Sea also creates a strong baroclinic circulation along the Eastern Side that brings Northern Atlantic water masses (green arrows) that eventually mix with fresher water masses to eventually create the Norwegian coastal current.

Strong local inflow of 0.29 Sv occurs via the straits of Pentland Firth and Faire Isle, and 1.33 Sv are obtained east of

the Shetland Islands (Thomas et al., 2005). Besides this, around 0.15 Sv come through the English Channel (Otto et al.,
1990; Thomas et al., 2005) and around 0.016 Sv are received from the Baltic Sea (Stigebrandt, 2001). Outflow to the North
Atlantic at the Northern end of the Norwegian Channel amounts to 1.79 Sv. Unlike the Baltic Sea, the North Sea is not
permanently stratified. During winter, enhanced wind induced mixing in combination with convective mixing maintain well
mixed conditions almost everywhere with the exception of the Norwegian Channel region where fresh outflow waters from
the Baltic Sea dominate (Schrum, 2001; Schrum et al., 2003; Ådlandsvik and Bentsen, 2007; Huthnance et al., 2009; Holt
et al., 2010; Mathis et al., 2013; Emeis et al., 2015; Mathis et al., 2015). During summer, the deeper parts of the central and
northern North Sea experience a strong thermal stratification (Mathis et al., 2013) whereas the shallow Southern North Sea
remains mostly well mixed around the year though short term stratified periods are possible, (Baretta-Bekker et al., 2009).
The timing and intensity of the seasonal stratification plays an important role for biogeochemical processes as it influences
primary production, the start of the spring phytoplankton bloom, and the nutrient cycling in the North Sea (Pätsch and Kühn,
2008; Holt et al., 2012; Daewel and Schrum, 2013; Gröger et al., 2013).

For both the North Sea (Mathis et al., 2013; Graham et al., 2017), and the Baltic Sea (Meier et al., 2003; Burchard et al.,
2009; Hordoir et al., 2015), there is a number of models of different complexity. Because the two seas differ so much in
their oceanographical and biogeochemical characteristics it is plausible to setup separate models specific for each region, and
to prescribe lateral boundaries at a reasonable location in the transition zone between the two seas. However, recent studies
provide evidence that the simulation of the Skagerrak and Kattegat hydrography are often problematic in these model setups
(Pätsch et al., 2017). As the dynamics in this transition zone between the Baltic and the North Sea are essential for a realistic
simulation of major Baltic Sea inflows (MBIs), a combined Baltic-North Sea model is necessary to better understand their
dynamics and their impact on the Baltic and North Sea physics and ecosystem. Especially for climate scenario simulations
it is more appropriate to explicitly simulate potential changes in the Kattegat/Skagerrak region, than relying on a present day
climatological prescription. Further, ocean models that include only the North Sea do not take into account the interaction with
the Baltic Sea, and therefore rely on a freshwater provision to the North Sea that is not accurate in strength nor in time (Hordoir
et al., 2013).
Early attempts of combined North Sea - Baltic Sea modelling were limited by a relative coarse resolution (approx. 11000
m, or 6 nm) and constrained on short term simulations of one year (Schrum and Backhaus, 2011). Only recently, attempts
have been done to include the two seas in one model setup for longer periods and in high resolution. Daewel and Schrum
(2013) used a coupled hydrodynamic-biogeochemical model and concluded that coupled Baltic and North Sea model setups
can help to improve the performance in the Skagerrak area. Moreover, the model reasonably simulated the major MBIs during
the hindcast period (Daewel and Schrum, 2013). Gräwe et al. (2015) presented a hydrodynamic model using adaptive vertical
coordinates and with a resolution of 1852 m (or 1 nm). The authors found the mean state for the period 1997-2012 as well
as the timing and amplitude of MBIs were well represented by the model. Tian et al. (2013) as well as Pham et al. (2014)
presented each a hydrodynamic model coupled to an atmospheric model but they focused more on atmospheric dynamics and
air sea interactions than on oceanographic topics.
In the present article, we provide a description and validation of Nemo-Nordic in its first mature version, Nemo-Nordic 1.0,
based on Nemo 3.6. Nemo-Nordic is declined into two resolutions of 2 nautical miles (3704 m) and 1 nautical mile (1852
m). First prototype versions were already used for process studies in (Hordoir et al., 2013, 2015) or to specifically investigate
the added value of interactive air-sea exchange of mass and energy fluxes (Gröger et al., 2015). The present version is the
first reference, used both for research and forecast. This article provides a full validation of both barotropic and baroclinic
dynamics, it shows the qualities and biases of the model. It is also a milestone that will be used for further development at
a time when Nemo-Nordic extends beyond SMHI in different European institutions within the Copernicus framework for
operational applications for example, but also for research.

Nemo-Nordic is a NEMO (Madec and the NEMO system team, 2015) based ocean model for Baltic & North Sea that can be
used for climate, oceanographic process study, and operational oceanographic applications. Nemo-Nordic has been designed to
be an advanced compromise to provide forecast or study Baltic and/or North Sea dynamics, at various time scales (operational
or climate time scales), with a representation of processes occurring in both basins including overflows and sea-ice, within a
reasonable range of computing resource. The inclusion of the both seas makes it possible to study the exchange between the
two seas because its boundary condition is far enough from where this exchange occurs, unlike for example a Baltic Sea only
ocean model (Meier et al., 2003). Unlike Nemo-Nordic, the model described by (Meier et al., 2003) is in any case limited by its
linear free surface, which produce conservation errors for the Baltic Sea in long term simulations, and forbids any possibility
of representing properly ocean dynamics in any region of higher sea level variability, such as Kattegat, Skagerrak and the entire
North Sea.
Nemo-Nordic is also not the first model to include both Baltic and North-Sea basins, one could cite (Funkquist and Kleine,

2007) or (Madsen et al., 2015) as examples of models who already do. However these models do not permit to represent the main driver of the Baltic Sea ecosystem, the dense overflows that feed its very specific sill bounded estuarine circulation: this circulation is difficult to represent in z coordinates since such coordinate systems do not represent well dense overflows. In addition, the Baltic Sea halocline is tilted, and is featured in a very low turbulence environment. Representing the Baltic Sea halocline therefore requires a possibility to rotate the diffusion tensor to avoid diapycnal mixing, and to limit the vertical mixing 105 length in case of low turbulence. The NEMO ocean engine allows to have tools such as bottom boundary layer parametrization, isopycnal diffusion in z coordinates, or advanced vertical turbulence schemes, that permit a better representation of such circulations. From a more general point of view, using a community engine such as NEMO means having the latest available developments in ocean and sea ice modelling. Further developments are now being made into Nemo-Nordic, such as wind wave coupling for example, that will keep this ocean modelling configuration to a state of the art level when it comes to Baltic 110 and North Sea modelling.

## 2 Model setup: Nemo-Nordic

Nemo-Nordic is an ocean model setup for Baltic & North Sea. It is based on the "Nemo ocean engine", a set of ocean modeling tools supported by a large community, and in constant watch towards developments done in other community ocean models. More specifically, we apply the stable NEMO 3.6 version (Madec and the NEMO system team, 2015). The ocean 115 component is coupled to the sea ice model LIM3 (Vancoppenolle et al., 2009). The first version of Nemo-Nordic were based on NEMO 3.3.1, switching to NEMO 3.6 which features a new coupling between barotropic and baroclinic modes ensures a much better representation of the barotropic mode, with a sea level representation of a higher quality. Technically, NEMO 3.6 and the use of the XIOS server was a key element to a more user friendly version of Nemo-Nordic.

### 2.1 Grid and bathymetry

The model domain of Nemo-Nordic covers the English Channel, the North Sea and the Baltic Sea (Fig. 3). This domain bears similarities with that used in the NEMO based configuration described by Maraldi et al. (2013). The resolution of the onfiguration of Maraldi et al. (2013) is higher than that of Nemo-Nordic, but also uses the features of the NEMO ocean engine to build regional and coastal configurations. The main interest of the two configurations is different: that of Maraldi et al. 125 (2013) aims at resolving the European shelf dynamics, whereas the purpose of Nemo-Nordic is to represent the interaction between two basins with different dynamical features.

The area of Nemo-Nordic reaches from 4.15278 W to 30.1802 E and 48.4917 N to 65.8914 N. The grid is geographical, and in its 2 nautical version the horizontal grid has zonal/meridional increments of 0.05°, which corresponds to a horizontal 130 resolution of approximately 2 nautical miles (3704 m). The resolutions of approximately 1 or 2 nautical miles are given as approximations, especially at these high latitudes where zonal scale factors differ between the Southern and Northern parts of the domain.

This resolution does not permit a proper description of the Danish Straits, which is a critical area of this configuration. However, in order to insure a proper communication between the Baltic and the North Sea, we tune a proper "impedance" of 135 the Danish Straits in the model so that the flow between the two areas is consistent. We never checked if this feature allows a proper representation of the baroclinic flow within the straits, but since major Baltic inflows are mostly barotropic events we believe this is of secondary importance. However a higher resolution of the Danish straits will be implemented in the future using the AGRIF tool.

Nemo-Nordic uses $z^*$ vertical coordinates, which is more simple than some other Baltic Sea models such as Gräwe et al. 140 (2015) or the configurations develloped by Shapiro et al. (2013). Advanced hybrid coordinates permit a better representation of dense overflows to the Baltic Sea, but have a higher computational cost. However, the salinity biases based on the latest advances made in Nemo-Nordic show very acceptable range, as shown in Section 4, and the benefit of $z^*$ coordinates from a vertical point of view allow multiple long term simulations with realistic computing power. Our model setup comprises 56 vertical levels. The vertical resolution is adapted to the physical properties of the Baltic and North Seas: the upper levels have 145 a thickness of approximately 3 m until the typical level of the halocline is reached at 60 m. Below 60 m the layer thickness increases substantially: the layer thickness is 10 m at a 100 m depth, which is the typical halocline depth of the Norwegian Coastal Current (NCC hereafter) (Skagseth et al., 2011). Maximum values of 22 m are reached below 200 m. Note that the maximum decrease in resolution between two consecutive vertical levels is below 11%. With these increments, the vertical resolution of Nemo-Nordic is adapted to Baltic Sea conditions and less optimal to resolve the halocline of the NCC. However

the NCC is a baroclinic Kelvin wave which has a permanent renewal of stratification coming from the constant runoff flux of the Baltic Sea and all the river inputs located along the Western Swedish and Norwegian coasts. Thus the NCC halocline is less sensitive to coarse vertical resolution than the halocline of the Baltic Sea, which is not permanently renewed and is suspected to impact spurious salt water inflows in ocean models. Partial steps are used at bottom level to ensure a proper fit between the input bathymetry and the vertical grid.

The cross sectional area of the Danish Straits is critical as it ensures the exchange between Baltic and North Seas. The resolution of the model does not permit to have a proper representation of the Danish Straits. However, it is possible to represent each cross-section so that it has the same hydraulic impedance as in reality. This was achieved by fitting the area of each critical cross-section of the model with its equivalent area in the real world. More precisely, this means that the area of each critical "numerical cross-section" of the model is made to fit with the surface of the real cross-sections of the Danish Straits.

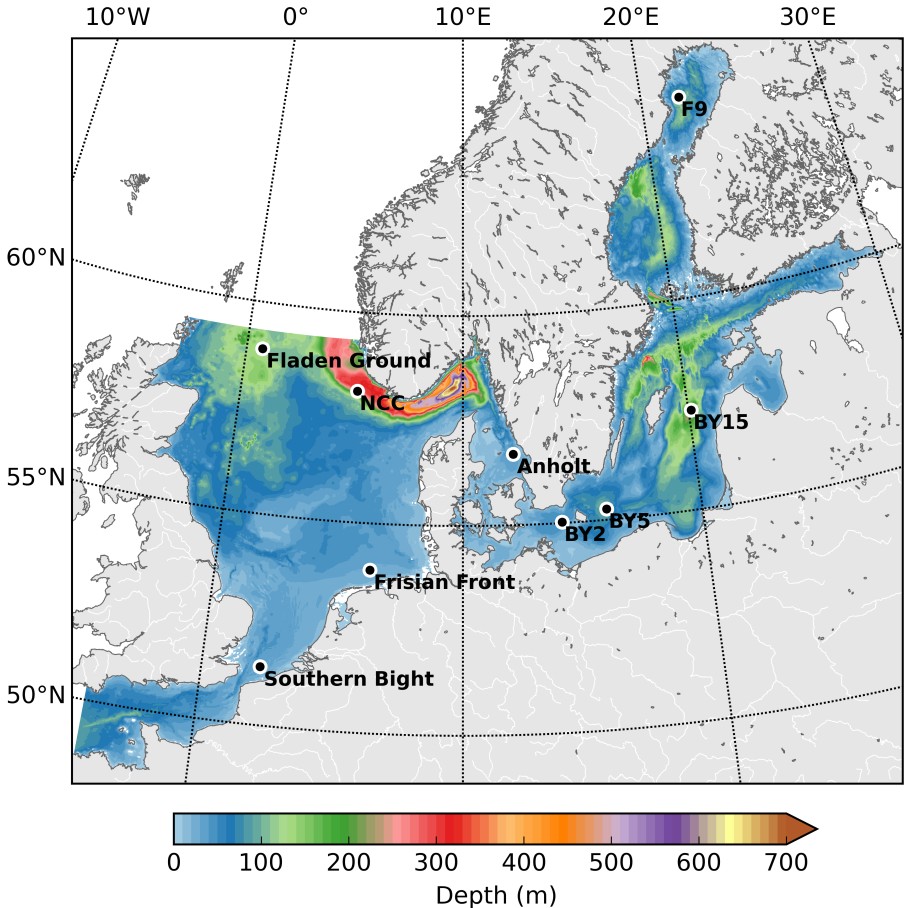

**Fig. 3.** The model domain and bathymetry of Nemo-Nordic. The filled circles show the locations of validation stations for salinity and temperature.

## 2.2 Physical Settings

### 2.2.1 Free Surface and Time Stepping

Nemo-Nordic uses a non-linear free surface (`key_vvl`) with a (`key_dynspg_ts`) option to split the computation of
barotropic and baroclinic modes. In comparisons with previous ocean models such as that used by Meier (2004) for example which used a linear free surface, these features enable Nemo-Nordic to run in regions of high tidal amplitude like the North Sea and the English Channel. Tuning is done in order to obtain an accurate SSH representation. We use a variable bottom roughness (`ln_bfr2d = .true.`) along the cyclonic pathway of barotropic Kelvin waves in the North Sea. This roughness is optimized to obtain the best possible SSH amplitude. Nemo-Nordic runs with a baroclinic time step of 360 s, which proved to be the best compromise between stability and computational speed. The barotropic time step is set to be 30 times smaller.

### 2.2.2 Mixing and Dense Overflows Representation

Vertical and horizontal mixing (and the connections between each of them) is a critical element when it comes to a Baltic & North Sea configuration. The major difficulty is to simulate the crucial dense water overflows in the Baltic Sea (Hordoir et al., 2015). In order to ease this process, Nemo-Nordic uses a bottom-boundary-layer parametrization (Beckmann and Döscher, 1997) for tracers (`key_trabbl`) in order to reduce the biases of a z-coordinate ocean models when it comes to the representation of dense overflows. In NEMO, the bottom-boundary-layer can be used with an upstream advection scheme or with a purely diffusive scheme, or with a mixture of both. For Nemo-Nordic, sensitivity experiments have shown that the use of a purely diffusive scheme results in more realistic deep salinity in the Baltic Sea. Using an advective scheme results in a lower deep salinity, most likely due to a higher entrainment rate during major Baltic inflow events. One could object rightly that using sigma coordinates could have solved this issue without the use of such a parameterization. But using such a coordinate system would on the other hand create a strong diapycnal mixing of the Baltic Sea halocline, and most likely pressure gradient errors in the steepest places of the North Sea such as the Norwegian trench.

However, the use of the bottom-boundary-layer parametrization is not enough to achieve a correct representation of the overflows. The tuning of the mixing was a major issue and required a tuning process. Nemo-Nordic includes two basins which have very different dynamical features: The North Sea is strongly influenced by tides, tidal mixing and relatively strong winds. In the North Sea, the stratification is relatively weak, with the exception of a summer thermocline in some regions, regions of freshwater influence (Simpson and Souza, 1995) and the region of the Norwegian Coastal current (Hordoir et al., 2013). In contrast, the Baltic Sea has almost no tides, is less exposed to wind forcing and has a strong permanent stratification. Nemo-Nordic uses a two equation turbulence closure, based on a $k - \epsilon$ turbulence scheme (`key_zdfgls`). In addition, the Galperin parameterization is used (Galperin et al., 1988), in order to preserve the Baltic Sea permanent stratification. This parameterization limits the mixing length computed by the vertical turbulence model in case of low turbulence and high stratification, which fits perfectly with the Baltic Sea halocline environment. This parameterization can be written following Equation 1, in which k is turbulent kinetic energy, N is the Brun-Vaisala, and $C_{galp}$ is the Galperin coefficient. Finally, $lmax$ is the maximum mixing length.

$$lmax = C_{galp} \frac{\sqrt{2k}}{N} \tag{1}$$

Experiments done on the configuration without this latest parameterization yield a Baltic Sea permanent halocline that vanishes in a few years only. The Galperin coefficient is set to 0.17 which proves to be a good compromise between deep salinity and seasonal thermal stratification. A too high Galperin coefficient produces a higher deep salinity, but also a very thin and un-realistic seasonal thermocline. In addition to the Galperin parameterization, the background levels of turbulent kinetic energy are turned to their minimum values, the goal being always to limit as much as possible mixing at the level of the Baltic Sea halocline.

Horizontal mixing is based on a Laplacian approach combined with the rotation of the horizontal diffusion tensor (`key_ldfslp`). However, this does not result in a real isopycnal diffusion in the sense that Nemo-Nordic is still based on a z coordinate system and not on isopycnal coordinates. Close to the bottom, sensitivity experiments have demonstrated that the rotation of the diffusion tensor did not enable to follow dense salt inflows: increasing horizontal diffusivity resulted in lower deep salinity, or even no penetration of dense inflows at all, showing that diapycnal mixing is created by horizontal diffusion, even when the rotation of the diffusion tensor is activated. The first versions of Nemo-Nordic used a Smagorinsky approach in order to limit horizontal mixing as much as possible, but this approach finally resulted in very weak Baltic Sea salt

inflows, and a diffusivity impossible to control. Our final strategy has therefore been to create a viscosity/diffusivity coefficient which is high where model stability requires it the most (i.e.: above the halocline), and low where it is crucial for diffusivity to be as low as possible to avoid diapycnal mixing. After a suite of sensitivity runs we decided on using spatially varying values for viscosity and diffusivity. From the surface to a depth of 30 m the viscosity is set to values ranging from 30 to 50 $m^2$ $s^{-1}$ for the Baltic and the North Sea respectively, while we set diffusivity to a tenth of this value. Below this depth the values of viscosity is reduced to 0.1 $m^2$ $s^{-1}$. The same increase was applied to the diffusivity which is still set to a tenth of the value for viscosity. This feature works as long as the Baltic Sea halocline is located at this precise depth, but should be changed if the halocline depth changes. Along the open boundaries, the viscosity is increased to 800 $m^2$ $s^{-1}$ over the entire water column, decreasing rapidly to standard values within a few grid points inside the domain. This value is huge but prevented the model to ever crash at the open boundary conditions. Several values were tried and it proved not have little effect on the sea level variability inside the model domain.

A free slip option is taken for the lateral boundary conditions.

### 2.3 Boundary Conditions

#### 2.3.1 Open Boundaries

Nemo-Nordic has two open boundaries, a meridional one in the western part of the English Channel, and a zonal one set between Scotland and Norway. The setting of the boundary conditions uses the open boundary condition module of NEMO (`key_bdy`), as well as the tide module (`key_tide`).

Several settings can be used for boundary conditions. Tidal harmonics are taken from Egbert and Erofeeva (2002); Egbert et al. (1994) although further investigation is also being made using the FES tidal model (Lyard et al., 2006). Harmonical values for SSH are interpolated on the open boundaries of Nemo-Nordic. Harmonical tidal transports (in $m^2$ $s^{-1}$) are also interpolated in the same manner and then divided by the local depth of the Nemo-Nordic domain. In research mode, temperature and salinity boundary conditions may come from climatological data or from climate simulations, and a simple storm surge model is used, model itself corrected by a global ORCA0.25 configuration to take into account seasonal variability due to temperature and salinity variations. In operational mode, Nemo-Nordic uses ECMWF forecasted data for its SSH, temperature and salinity boundary conditions.

#### 2.3.2 Atmospheric forcing

Nemo-Nordic has been so far used in forced mode using prescribed atmospheric conditions and the CORE bulk formulation (Large and Yeager, 2009).

To cover a long period, the atmospheric forcing is based on different sources. The driving data of the period 1961-1978 is based on downscaled ERA40 data (Uppala et al., 2005). The ERA40 reanalysis is downscaled with the Rossby Centre regional atmospheric model version 4 (RCA4) with spectral nudging (Berg et al., 2013). The downscaling is necessary to improve the horizontal resolution of the global reanalysis data set. Here, we use data from an RCA4 set-up with a horizontal resolution of 11km. The frequency of the driving data is hourly.

With the beginning of 1979 we change the atmospheric forcing to the SMHI reanalysis product EURO4M which is available until the end of 2013 (Dahlgren et al., 2016; Landelius et al., 2016). The EURO4M data is available 3 hourly and with a horizontal resolution of 22km. EURO4M incorporates data assimilation which assures the best quality for our driving data. In operational mode (Nemo-Nordic 1nm), the simulations were forced by HIRLAM C11 (hirlam.org).

In forecast mode, Nemo-Nordic uses a combination of hourly ECMWF LL01 (9 km) data and Arome-data (2.5km).

#### 2.3.3 Light Penetration

A proper light penetration parameterization proved to be an important feature in order to reproduce the proper thermal structure, and especially the formation of a summer "Cold Intermediate Layer". The Baltic Sea has turbid waters which prevent deep light penetration, concentrating the summer heat input close to the surface, and hence easing the autumn cooling. Nemo-Nordic uses a Red-Green-Blue light penetration parameterization, together with a constant chlorophyll value of 0.5 mg $m^{-3}$ to represent the turbid Baltic Sea waters. Failing to use chlorophyll concentration results in a too thin and too shallow cold intermediate layer. This chlorophyll concentration does not pretend to be realistic from a biogeochemical point of view, it corresponds to a

**Table 1.** SSH representation, in terms of correlation, standard deviation (meters) and root-mean-square-deviation (meters), made by Nemo-Nordic for 9 Baltic Sea stations. The comparison is based on a 18 months period, starting on July 1st 2011, and on hourly output SSH.

|            | Std. Obs | 1nm std | Corr. 1nm | RMSE 1nm | Std. 2nm | Corr. 2nm | RMSE 2nm |
|------------|----------|---------|-----------|----------|----------|-----------|----------|
| Kalix      | 0.28     | 0.27    | 0.97      | 0.07     | 0.24     | 0.96      | 0.076    |
| Furögrund  | 0.24     | 0.23    | 0.96      | 0.07     | 0.21     | 0.95      | 0.075    |
| Spikarna   | 0.2      | 0.2     | 0.95      | 0.06     | 0.18     | 0.94      | 0.07     |
| Forsmark   | 0.19     | 0.19    | 0.95      | 0.06     | 0.17     | 0.94      | 0.07     |
| Landsort   | 0.18     | 0.18    | 0.95      | 0.06     | 0.16     | 0.95      | 0.07     |
| Kronstadt  | 0.29     | 0.29    | 0.95      | 0.09     | 0.27     | 0.95      | 0.09     |
| Öland      | 0.17     | 0.17    | 0.95      | 0.06     | 0.16     | 0.94      | 0.06     |
| Simrishamn | 0.19     | 0.19    | 0.94      | 0.06     | 0.18     | 0.94      | 0.06     |
| Skanor     | 0.2      | 0.20    | 0.92      | 0.08     | 0.19     | 0.92      | 0.08     |

mean value for both basins which allow a realistic light penetration in an area which has a water turbidity higher than that of the global ocean.

### 2.3.4   River discharge

The Baltic Sea salinity is sensitive to the accumulated freshwater input since the exchange with the open ocean is very limited (e.g. Meier and Kauker, 2003). Freshwater supply to the Baltic Sea in the simulation must therefore be handled with care. Here, we are using data from the HYdrological Predictions for the Environment (HYPE) model (Donnelly et al., 2016). The model simulates a mean runoff to the Baltic Sea of roughly 16000 m$^3$ s$^{-1}$ for the period 1981-1998. However, Meier and Kauker (2003) state that the observed runoff for the same period is 15053 m$^3$ s$^{-1}$ with an even lower value (14085 m$^3$ s$^{-1}$) for the period 1902-1998. Consequently, we reduced the freshwater supply computed by HYPE. We have chosen a general reduction of 10% which gives more realistic runoff for the Baltic Sea. Moreover, this improved the Baltic Sea salinity (not shown). The freshwater input to the North Sea including the Skagerrak and Kattegat area amounts to 11515 m$^3$ s$^{-1}$ after the reduction by 10%. The river runoff is spread over 424 river mouths in the entire model domain whereas more than 250 are located in the Baltic Sea (excluding the Skagerrak and Kattegat area). The UNESCO equation of state for sea water is used. The reference density is set to 1035 kg m$^{-3}$, which is most likely over estimated. Further experiments should be done on this point.

### 2.3.5   Sea-Ice

Nemo-Nordic benefits from the use of the Nemo ocean engine and of its advanced sea ice model LIM3. Sea ice is a Baltic Sea specific feature that owes to the Baltic Sea low salinity. Being able to represent sea ice dynamics properly is compulsory when it comes to a Baltic Sea ocean model. Models such as Gräwe et al. (2015) do not include this feature. The sea ice in the Nemo-Nordic is validated in Pemberton et al. (2017). Comparison done with (Funkquist and Kleine, 2007) suggest that Nemo-Nordic reaches so far the highest accuracy of sea ice representation for the Baltic Sea.

## 3   Validation of barotropic mode and surface currents

### 3.1   Sea Level

To model and forecast SSH is one of the major aims of Nemo-Nordic. This section provides a statistical comparison between measured and modeled SSH at different tide gauges in the Baltic & North Seas.

We have chosen tide gauges which are as representative as possible for three respective regions: The Baltic Sea (Table 1), The Danish Straits plus Kattegat and Skagerrak (Table 2), and the North Sea, including the English Channel (Table 3). The comparisons, are based on a 18 month period, lasting from July 1st 2011 and ends on December 31st 2012 on an hourly frequency. The respective model simulation was started one month before to allow for a spinup time. Each area is presented in a specific array.

There is generally a very high correlations between model and observations in almost all regions. In the North Sea and the English Channel, correlations are highest and mostly close to 0.99, with the one exception of Hanstholm, where correlations are around 0.93. In the Baltic Sea, correlations are always close to 0.95. In the narrow Danish Straits and the Kattegat, regions

**Table 2.** SSH representation, in terms of correlation, standard deviation (meters) and root-mean-square-deviation (meters), made by Nemo-Nordic for 9 stations located in the Danish Straits, Kattegat and Skaggerak. The comparison is based on a 18 months period, starting on July 1st 2011, and on hourly output SSH.

|            | Std. Obs | 1nm std | Corr. 1nm | RMSE 1nm | Std. 2nm | Corr. 2nm | RMSE 2nm |
| ---------- | -------- | ------- | --------- | -------- | -------- | --------- | -------- |
| Gedser     | 0.23     | 0.21    | 0.93      | 0.08     | 0.21     | 0.94      | 0.08     |
| Barsebäck  | 0.18     | 0.19    | 0.88      | 0.09     | 0.14     | 0.68      | 0.13     |
| Klagshamn  | 0.18     | 0.18    | 0.91      | 0.08     | 0.19     | 0.92      | 0.08     |
| Rödvig     | 0.21     | 0.2     | 0.93      | 0.08     | 0.19     | 0.92      | 0.08     |
| Göteborg   | 0.22     | 0.21    | 0.91      | 0.09     | 0.18     | 0.93      | 0.08     |
| Smögen     | 0.22     | 0.23    | 0.91      | 0.1      | 0.2      | 0.93      | 0.08     |
| Viken      | 0.21     | 0.19    | 0.90      | 0.1      | 0.18     | 0.90      | 0.1      |
| Skagen     | 0.24     | 0.26    | 0.90      | 0.11     | 0.225    | 0.91      | 0.1      |
| Kungsvik   | 0.24     | 0.25    | 0.92      | 0.1      | 0.22     | 0.93      | 0.09     |

**Table 3.** SSH representation, in terms of correlation, standard deviation (meters) and root-mean-square-deviation (meters), made by Nemo-Nordic for 6 stations located in the North Sea. The comparison is based on a 18 months period, starting on July 1st 2011, and on hourly output SSH.

|           | Std. Obs | 1nm std | Corr. 1nm | RMSE 1nm | Std. 2nm | Corr. 2nm | RMSE 2nm |
| --------- | -------- | ------- | --------- | -------- | -------- | --------- | -------- |
| Calais    | 1.95     | 1.5     | 0.99      | 0.55     | 1.65     | 0.98      | 0.4      |
| Dover     | 1.8      | 1.4     | 0.99      | 0.4      | 1.5      | 0.99      | 0.3      |
| Aberdeen  | 1.05     | 0.95    | 0.99      | 0.15     | 0.95     | 0.99      | 0.15     |
| Plymouth  | 1.35     | 1.25    | 0.98      | 0.25     | 1.3      | 0.98      | 0.25     |
| Helgoland | 0.9      | 0.74    | 0.98      | 0.2      | 0.82     | 0.97      | 0.2      |
| Hanstholm | 0.28     | 0.24    | 0.93      | 0.11     | 0.26     | 0.98      | 0.1      |

with complicated topography, the correlations are lowest, but still exceed generally 0.9. Exceptionally low correlations (0.68) are obtained in Barseback and only when Nemo-Nordic 2nm is used. When Nemo-Nordic 1nm is used, the minimum depth of the ocean model is set to 3 m, compared to 9 m, which gives a better representation of the shallow banks located in the Öresund area, and of the amplification of barotropic waves. This feature was not implemented in Nemo-Nordic 2nm where the main focus is the Baltic / North Sea exchange. Using wetting and drying in the future should help better representations of SSH in shallow areas affected by strong sea level variability.

Biases in terms of SSH representation can be summarised as follows. Nemo-Nordic usually has a negative bias in terms of representation of the low frequencies in the North Sea, Skagerrak and Kattegat. In the same regions, the opposite bias can be noticed when it comes to higher frequencies: tidally driven SSH can present overshoots in some places. In the Baltic Sea, one can notice that the tidal signal is usually a bit too high, but since its amplitude remains very small in comparison of other frequencies it does not affect the quality of the SSH representation significantly. The representation of lower frequencies does not reveal any significant bias in the Baltic Sea.

Nemo-Nordic 1nm represents SSH better than Nemo-Nordic 2nm, which partially is due to the higher resolution for most features. For example in the Danish Straits, the representation of shallow banks has been taken into account in Nemo-Nordic 1nm, and the size of the cross sections has been adapted to represent the North Sea / Baltic Sea exchange. This allows to amplify incoming barotropic waves which is essential in order to get the right SSH variability in the Danish Straits tide gauges. In Nemo-Nordic 2nm, the main concern has been to insure the right Baltic / North Sea exchange which is crucial for having a correct representation of Baltic Sea salt inflows, which are one of the main drivers of the Baltic Sea ecosystem, and of its long term thermo-haline structure.

In summary, we identified the following key processes to model realistic SSH variations:

– In Nemo-Nordic, the SSH and SSH variability in the North Sea is to a first order barotropicaly driven, and is built by a combination of tidal waves entering through the Northern boundary and Western Boundaries, wind driven SSH built over the Northern Atlantic, and wind driven SSH over the North Sea. To get a correct representation of the SSH variability and the cyclonic circulation in the North Sea, it is important to have high frequency (hourly) boundary conditions that takes into account all these aspects.

– In the Kattegat/Skagerrak region, as one moves further towards the entrance of the Baltic Sea, the effect of tides and of high

frequency waves generated in the Northern Atlantic or the North Sea becomes less important. The low frequencies, on the other hand, generated by the storm surge model over the North Atlantic turned out to be important for this region. The shelf break along the coast amplifies the effect of coasts on barotropic waves arriving from the North Sea and helps representing the SSH variability and its extremes. A higher vertical resolution in the shallow areas improves the representation of the SSH variability, especially in the Skagerrak-Kattegat. This last effect becomes crucial in the Danish Straits where the shallow banks need to be represented.

– The SSH variability in the Baltic Sea is barely influenced by any tidal variability, but is highly influenced by low frequency forcing coming from the Northern Atlantic and the North Sea. In addition, local wind forcing over the Baltic Sea explains higher frequencies. The only communication between the Baltic and the North Sea being the Danish Straits, the adjustment of cross-sections and of the friction in this area are of crucial importance to chose which barotropic frequencies can penetrate the Baltic Sea. The Danish Straits should act as a well tuned low pass filter which allows low frequency waves to penetrate the Baltic Sea, but lets little high frequency power enter the Baltic Sea.

## 3.2 Surface Currents

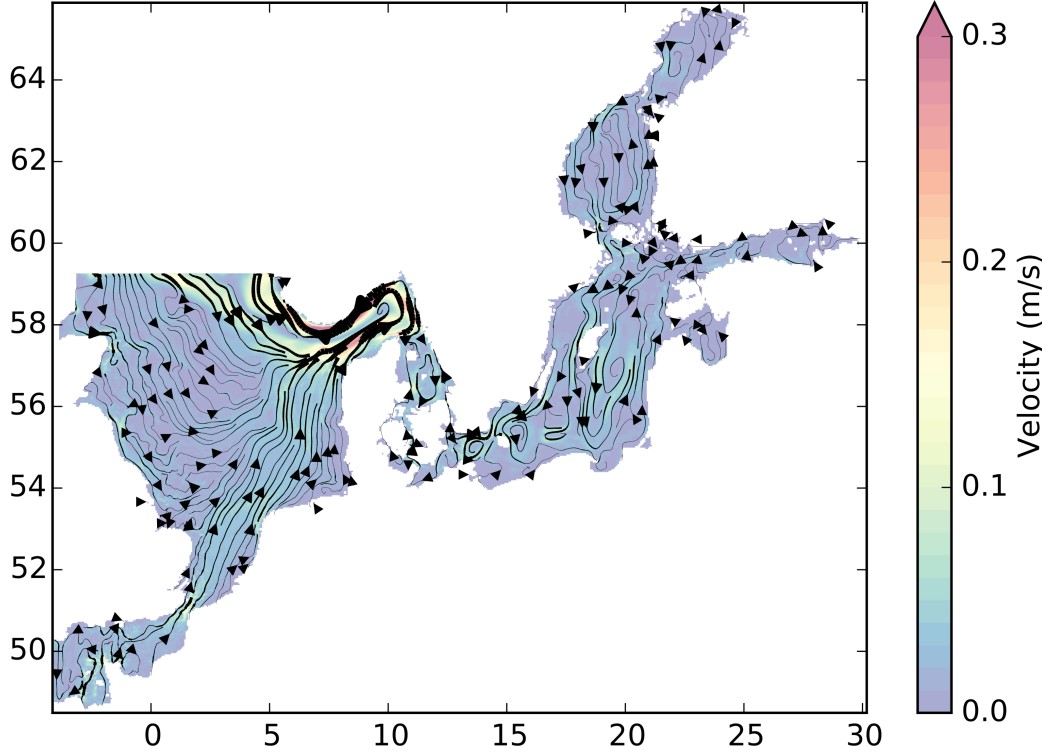

**Fig. 4.** Simulated surface (0-30 m) currents, climatology for 1979-2010. The lines and arrows show the streamlines and directions of the current vector field. The thickness of the line is scaled with respect to the speed of the current. The filled contours show the current speed in m/s

### 3.2.1 General circulation

The model reproduces the general cyclonic surface circulation pattern in the North Sea (OSPAR, 2000), with a southward flow in the western part of the basin, a northeastward flow along the southern coast, and a northward flow along the Norwegian coast in the Norwegian coastal current (Fig. 4). The strongest modelled southward flow of Atlantic water through the Northern boundary occurs just next to the NCC. This flow forms a current that flows South-East and enters the well known cyclonic circulation pattern in the Skagerrak, in good agreement with the observed surface currents in the North Sea (OSPAR, 2000). A part of the southward flow along the British Isles also deviates eastward to directly join the eastward flow towards the Skagerrak. Another part flows further to the South and mixes with inflow from the English channel. The larger part of inflowing Atlantic

water from the Northern boundary is restricted to north of 54 °N and then recirculates mainly following the topography of the Dogger Bank. The southern North Sea is dominated by inflows waters from the English Channel.

In the Baltic Sea and its sub-basins the model reproduces the observed general cyclonic circulation patterns (Elken and Matthäus, 2008), with a Southward flow in its Western part, and a Northward flow in its Eastern part. In the Gotland basin the model reproduces the Southward flow on both sides of Gotland, and the Northward flow along the coast of Baltic, giving rise to the cyclonic structure over the Gotland deep (Fig. 4). In the Kattegat the model simulates a general anticyclonic flow in agreement with Nielsen (2005), and resolves the northward flowing Baltic Current along the Swedish coast, that feeds low saline waters into the NCC.

### 3.3 Overturning circulation

The inflows and outflows through critical cross-sections, where observational estimates exists, have been calculated for the period 1979-2010. Inflows are defined as volume transports directed inwards to the domain, and outflows are defined as transports directed outwards. For the Baltic Sea a longitudinal cross section has been taken along 12.90 °E, and inflowing waters are defined as transports in the positive x-direction, and outflows in the negative. For the strait of Dover a longitudinal cross-section has been taken along 50.99 °N. Here the flow is barotropic and there is only a mean inflow to the North Sea. For the Northern Boundary, taken along 58.06 °N, the inflow is all transports in the negative y-direction, and the outflow is all transports in the positive y-direction.

As shown in table 4, the modelled transports agrees well with observational estimates, which is an additional indication that the general circulation of the North and the Baltic Sea is well reproduced by the model. The modelled flows gives a water residence time of 23 and 1.8 years in the Baltic Sea and the North Sea, respectively.

The circulation in the North Sea is mainly of a barotropic feature. Barotropic stream functions of the horizontal overturning circulation, showing the general cyclonic circulation of the North Sea, are displayed in Figure 5. The largest flows are found in the Norwegian trench, where the overturning barotropic circulation amounts to 0.9 Sv, in good agreement with the calculated fluxes in Table 4

**Table 4.** Volume flow (Sv) through cross-sections, climatological mean for 1979-2010.

| Cross section | Model | | Observations | | Reference |
|---|---|---|---|---|---|
| | Inflow | Outflow | Inflow | Outflow | |
| Strait of Dover | 0.110 | - | 0.11-0.17 | - | Otto et al. (1990), and references therein |
| Northern boundary | 0.794 | 0.928 | 0.93-1.73 | 1.34-1.8 | Otto et al. (1990), and references therein |
| Baltic Sea | 0.029 | 0.044 | 0.027 | 0.043 | Savchuk (2005) |

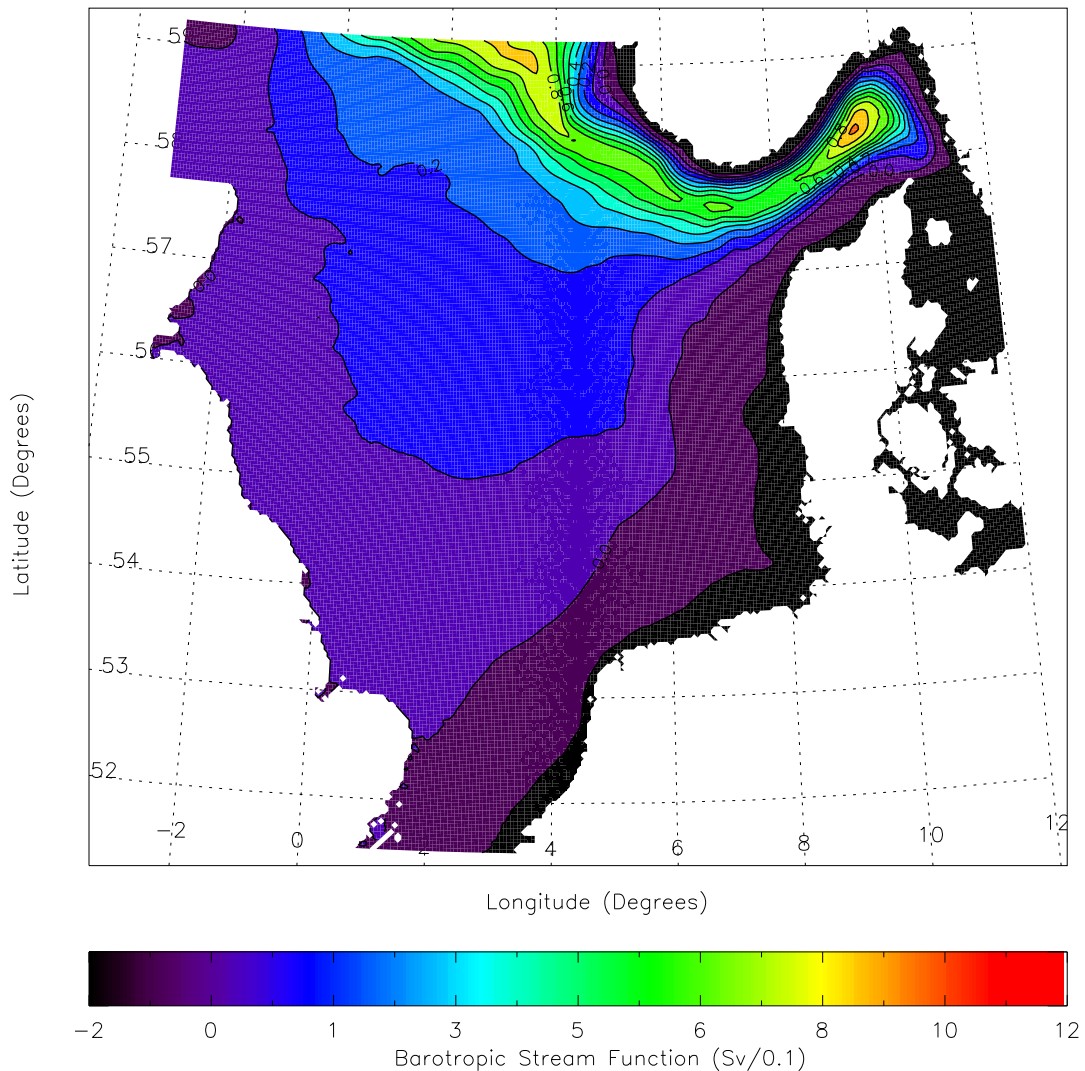

**Fig. 5.** Barotropic stream function (Sv), climatology for 1979-2010.

## 4   Salinity and temperature

### 4.1   Surface temperature and salinity

A validation of the simulated mean surface salinity and temperature is made (Figures 6 and 7) using the Janssen et al. (1999) climatology. The values are computed as a mean value over the first ten meters. For the Baltic and North Sea, the overall surface salinity is well reproduced. There is, however, a positive surface bias in the Baltic Sea. Especially one can notice that the penetration of the 8 PSU iso-haline within the estuary is too high. The 7 PSU iso-haline is also located a few nautical miles too far North. In the North Sea there is a negative bias in freshwater-influenced areas (the NCC and in the Southern
German Bight). For Sea Surface Temperature (SST), the structure is similar to observations but there is a positive bias of less than 1 degree over all the domain. This bias seems to partially come from a warm bias in the atmopheric forcing (in winter) (Landelius et al., 2016), and partially from surface overshoots during the summer period. Sensitivity experiments have shown that the positive SST bias during summer time is lower if the Galperin coefficient used to maintain a stable haline stratification is lowered. In order to avoid these effects, further development is being made to decouple the Galperin parameterisation from
thermal effects.

### 4.2   Thermohaline structure of the Baltic Sea

The thermohaline structure of the Baltic Sea exhibits two types of variability. First, a spatial variability which makes that the Baltic Sea has strong salinity gradients from the surface towards the bottom, but also presents estuarine features which results in a decreasing salinity from South towards North. Strong temperature gradients also exist as the Northern regions of the Baltic
Sea have a much colder climate than those located in its Southern part. From a temporal point of view, surface salinity exhibits a seasonal variability (Hordoir and Meier, 2010), and deep salinity has a lower variability highly related to the occurrence of deep salt inflows (Hordoir et al., 2015). Surface temperature has a strong seasonal cycle related with summer stratification and its destruction during autumn. The models ability to reproduce the thermohaline structure and its variability on different time scales will be validated below.

### 4.2.1   Vertical structure and seasonal variability

The haline vertical structure of the Baltic Sea is in general well reproduced by Nemo-Nordic (Fig. 8), with a distinct halocline separating the surface waters from the deeper waters. In the Bothnian Bay and the Kattegat (F9-A13 and Anholt), the modeled depth (50 and 20 meters, respectively) and strength of the halocline, corresponds well to observations. There is a small negative salinity bias over the whole water column in the Bothnian Bay, while the overall salinity is well reproduced in the Kattegat. In
the Baltic Proper and the Bornholm Basin (BY15 and BY5), the modeled halocline is weaker and shallower than the observed one. The surface water at these stations tend to have a positive bias, while the deeper waters tend to have a negative bias, suggesting a too strong mixing between surface and deep waters. No stronger seasonal cycles exist in the haline structure, except for a freshwater pulse arriving in the surface waters during summer months, which is captured by the model at all stations.

The thermal vertical structure is dominated by the seasonal thermocline. In the Kattegat, Bornholm Basin and the Baltic proper (Anholt, BY5 and BY15), it starts forming earlier than in the Bothnian Bay (F9-A13), both in the model and in the observations (Fig. 9). The later formation of the thermocline in the Bothnian Bay is partially due to the lower insolation at higher latitudes, but also due to the non-linearity of the equation of state at low salinities. The temperature of maximum density is higher at lower salinities, meaning that the water column has to completely mix before the formation of the thermocline can
start, which is well reproduced by the model. The model's development of the seasonal thermocline and its deepening agrees well with observations in the Kattegat, the Bornholm basin and in the Baltic proper. In the Bothnian Bay no conclusions can be drawn on this aspect due to a lack of measurements between 15 and 40 meters, where the thermocline is situated. The break up of the thermocline is also well represented in the model. The model has on the other hand difficulties in representing the colder intermediate waters in the Bornholm basin and the Baltic Proper, which have a warm bias. This might be related to biases in
the atmospheric forcing (Landelius et al., 2016), as these waters are formed during winter convection. Indeed, the simulated winter surface temperatures at BY5 and BY15 tend to have a warm bias. The modelled cold intermediate waters also descend too deep towards the end of the year. This might be related to the weaker halocline, and thus probably stronger mixing, in the model. The deep waters below the halocline are warmer than the intermediate waters, which is reproduced by the model. In the model it is, however, about 0.5 degrees warmer than in the observations at the BY15 station.

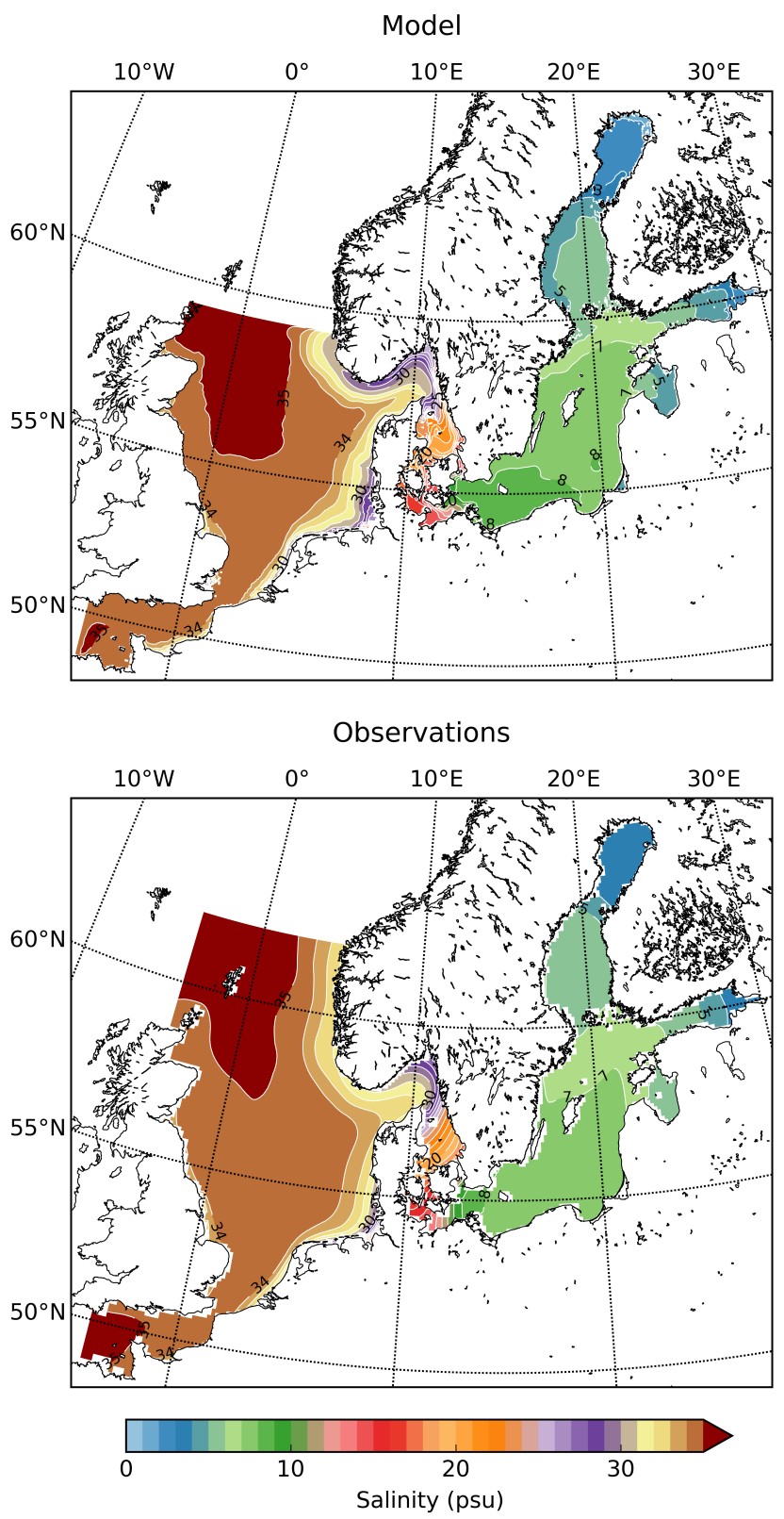

**Fig. 6.** Mean surface (0-10 m) salinity for Baltic, North Sea and English Channel, as simulated by Nemo-Nordic for the period 1979-2010 (upper figure) and from observations (lower figure) from Janssen et al. (1999).

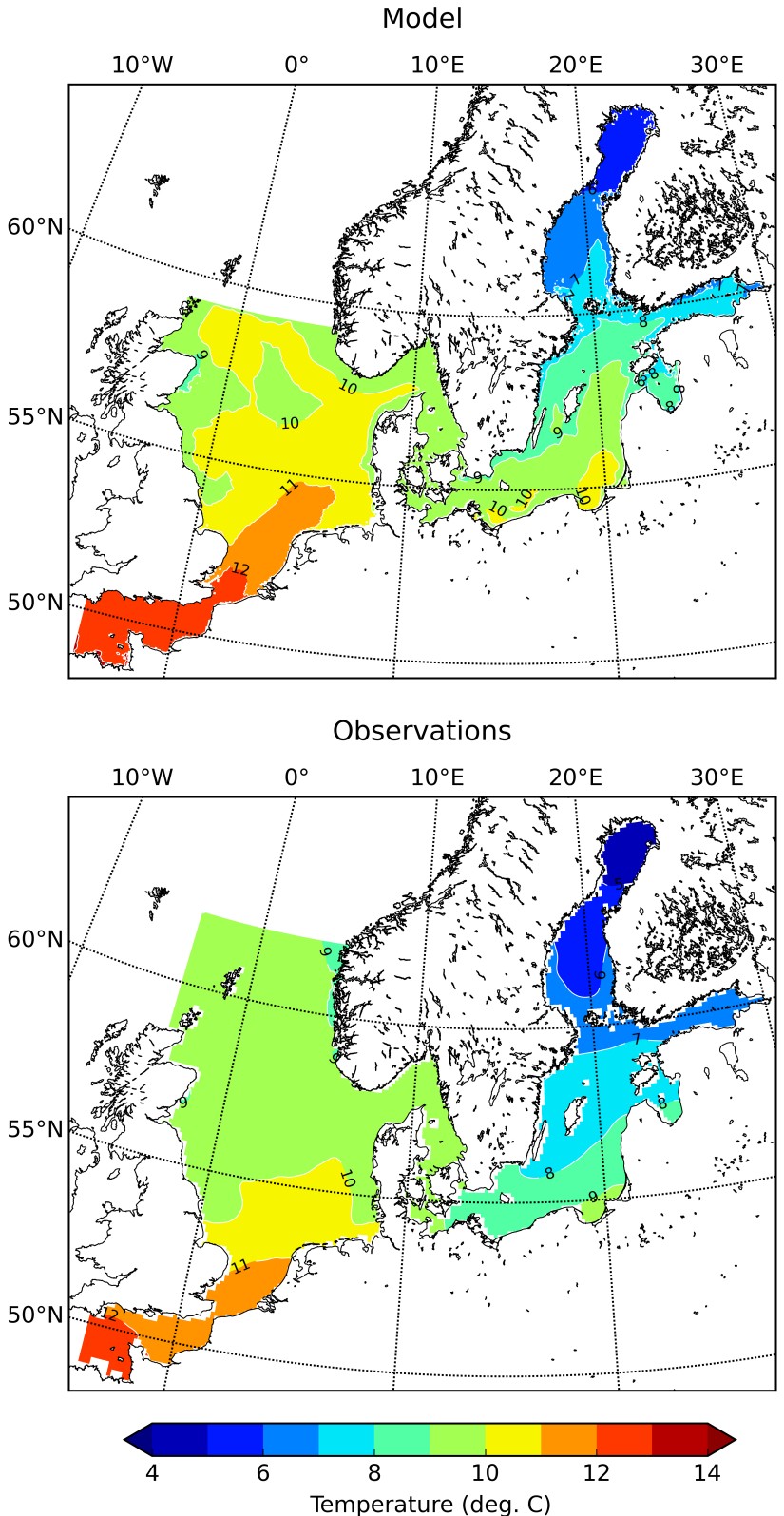

**Fig. 7.** Mean surface (0-10 m) temperature for Baltic, North Sea and English Channel, as simulated by Nemo-Nordic for the period 1979-2010 (upper figure) and from observations (lower figure) from Janssen et al. (1999).

## Salinity (psu)

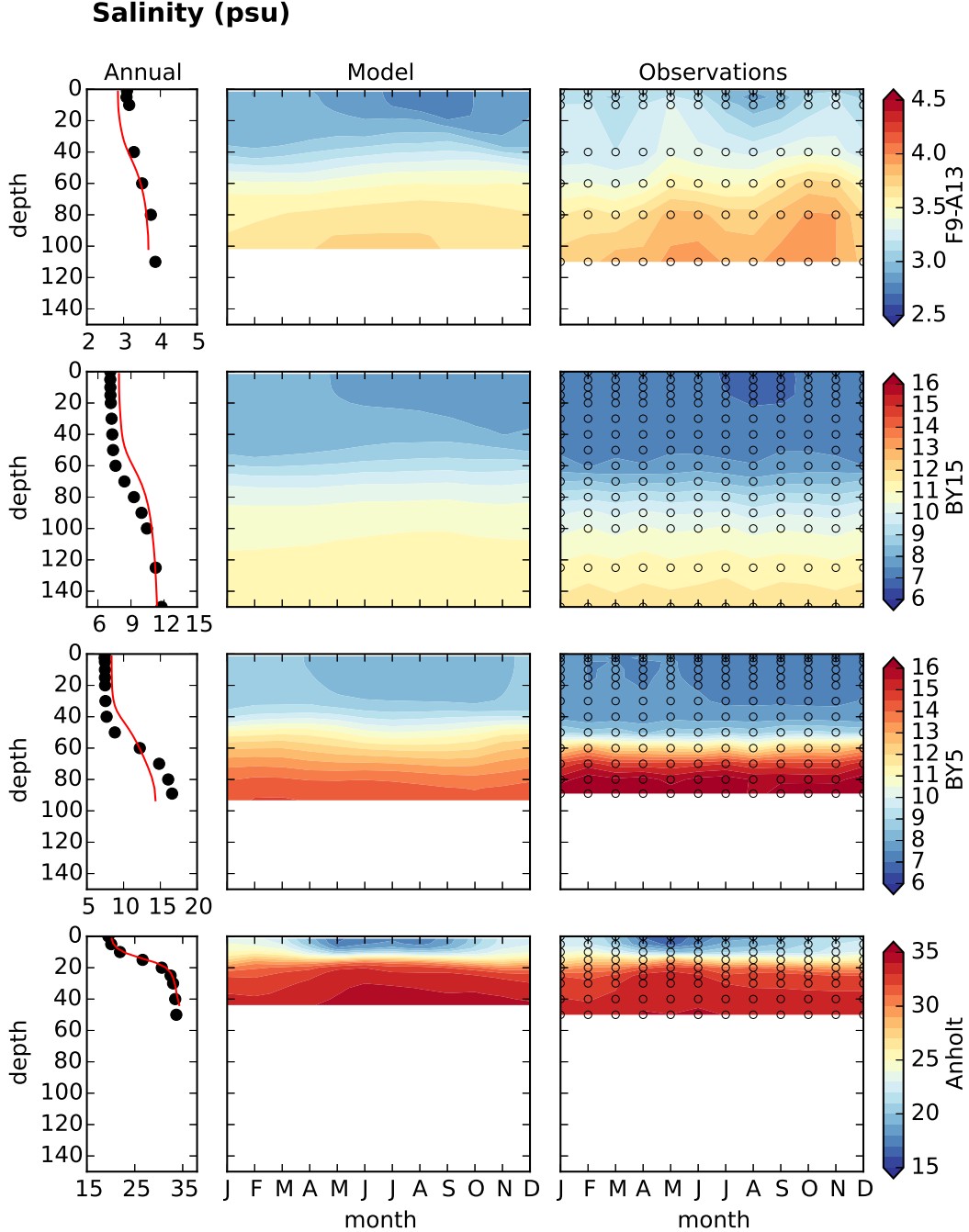

**Fig. 8.** Haline structure for four different stations of the Baltic Sea, from top to bottom F9-A13 station in the Bothnian Bay, BY14 station in the Baltic Proper, BY5 station in the Bornholm Basin and Anholt station in Kattegat. The left column displays annual mean depth profiles, where the red line is simulated salinity and black dots are observations. The middle and the right columns show seasonal variations in the model and the observations, respectively. In the right column the transparent circles show sample depths, the observations are based on a 1979-2010 climatology.

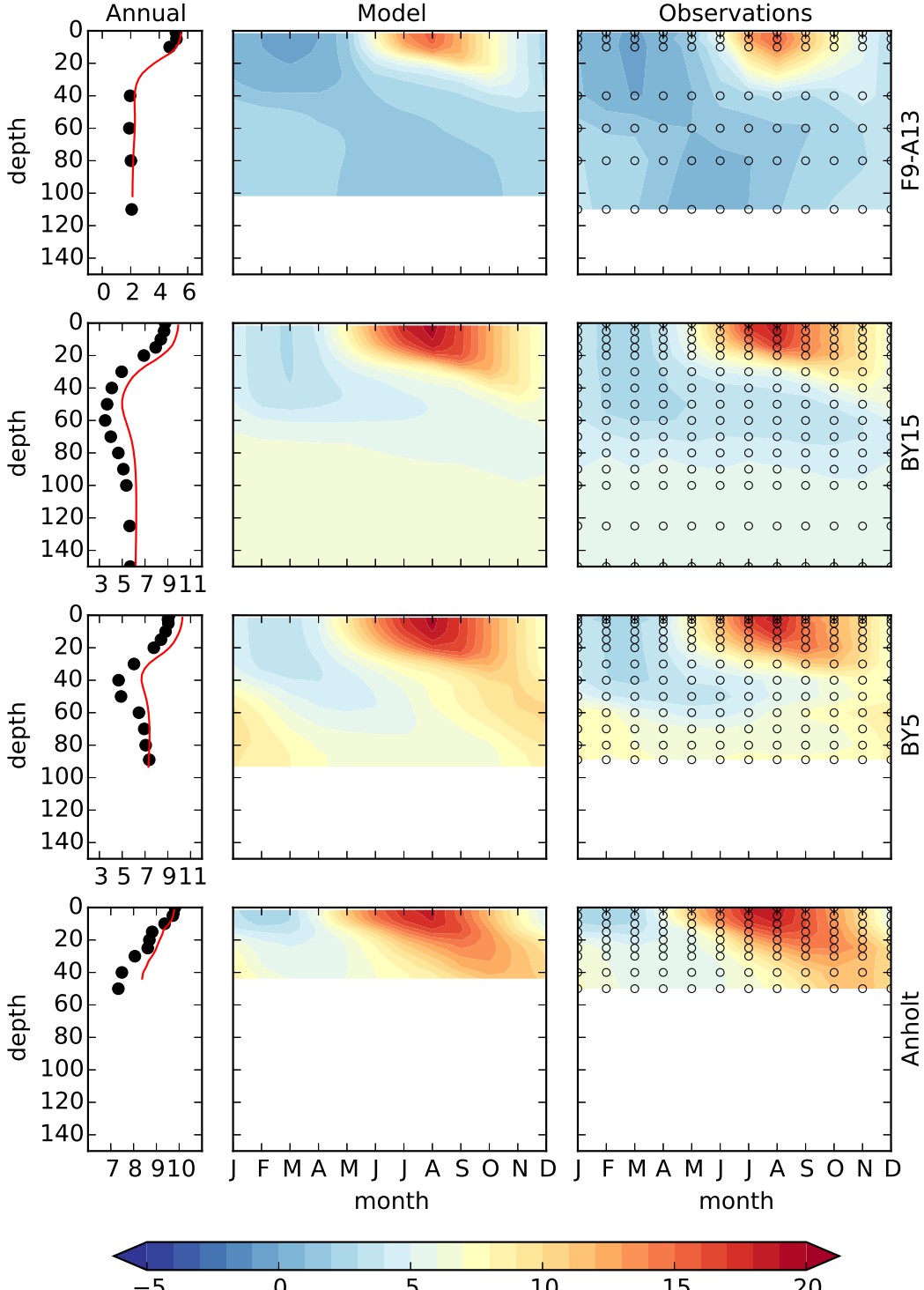

**Fig. 9.** Thermal structure for four different stations of the Baltic Sea, from top to bottom F9-A13 station in the Bothnian Bay, BY14 station in the Baltic Proper, BY5 station in the Bornholm Basin and Anholt station in Kattegat. The left column displays annual mean depth profiles, where the red line is simulated temperature and black dots are observations. The middle and the right columns show seasonal variations in the model and the observations, respectively. In the right column the transparent circles show sample depths, the observations are based on a 1979-2010 climatology.

### 4.2.2  Interannual variability

From the comparison between the observations and their variability (Figure 10), one notices that Nemo-Nordic in general well reproduces the variability in the deep water salinity close to the bottom, both in the Kattegat and in the Baltic Sea, despite the constant background bias in the salinity. Especially, it is interesting to note from the BY15 salinity that the model is able to reproduce the Major Baltic Inflows of 1993 and 2003. The mechanisms behind the Major Baltic Inflows are actually mostly barotropic, although their propagation to the bottom of the Baltic Sea involves mixing processes. Therefore the representation of sea level variability is a key element into reproducing the variability of the Major Baltic Inflows. A comparison of the modelled and measured sea level differences during the recorded durations these inflows shows the models accuracy to represent the underlying barotropic processes. During the duration of the 1993 major Baltic inflow, the sea level at Landsort increases by 1 m in observations against 1.02 m in the model, while during the 2003 major Baltic inflow, the sea level at Landsort increases by 0.58 m both in the model and in the observations. This further suggests that a miss-represented baroclinic process must be responsible for the negative bias in deep water salinity in the Baltic Sea. Even though this model uses a bottom boundary layer parameterisation (Beckmann and Döscher, 1997), it is still a z coordinate model which makes that the representation of dense overflows is not as accurate as it would be in a sigma or generalized vertical coordinate model. It is also interesting to note that Nemo-Nordic tends to overestimate the variability in deep water salinity at the Anholt station in the Kattegat, while it slightly underestimates the variability at the BY2 station.

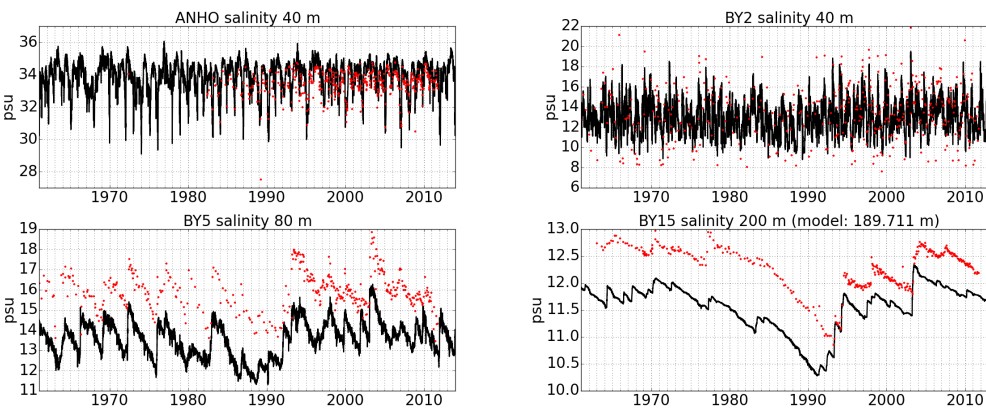

**Fig. 10.** Time series of modeled salinity (black) and observations (red) for different stations. The stations are from top to bottom Anholt, BY2, BY5 and BY15 (check Fig 3 for their location). All stations are relevant for inflowing salty water masses from the North Sea to the Baltic Sea. The chosen levels are all close to the bottom of the corresponding location.

### 4.2.3  Short term variability - Comparison with Argo Floats Data

As an example of Nemo-Nordic's performance on shorter term, we compared the model results to temperature observations from an autonomous Argo buoy. Over 100 profiles were collected during a mission in the Bothnian Sea, which lasted from 13 Jun 2013 to 2 Oct 2013. The profiles were taken from between $61.57°$N and $62.47°$N in latitude and between $19.59°$E and $20.42°$E in longitude. This dataset has been described in detail by (Westerlund and Tuomi, 2016), who also provided an illustration of the buoy route. From a comparison of observed and modelled temperature profiles we see that the seasonal thermocline is visible throughout the summer (Figure 11). The vertical structure of temperature was relatively well reproduced by Nemo-Nordic near the surface. In August the thermocline reached maximum depth and temperature. The model was able to describe well how the mixed layer deepened during the summer. The temperature gradient of the thermocline was also well represented. The surface layer responded to atmospheric forcing in a similar way in observations and model. In layers under the thermocline, model temperatures were somewhat too high. This bias increased in late summer. Deeper, the dicothermal (old winter water) layer was not as pronounced in the model as it was in the observations. In late August, model predicted larger thermocline depths than were observed. In general, temperature profiles were smoother in the model than in observations. Furthermore, some finer scale features were not completely reproduced by the model.

Observed and modelled near-surface temperatures, along with estimated thermocline depths, are shown in Figure 12. Near-surface temperature was taken to be the temperature of the model point at the depth of the topmost data point in the observations, which was typically around 4 metres, depending on the profile. In most cases this is very close to the surface temperature. Location of the thermocline was taken as the place of the maximum temperature gradient along the z-axis. Near-surface

temperature in the model reproduced overall seasonal temperature cycle, although in early September surface temperatures
were around 1 degree greater in the model than in observations. Thermocline depths were represented in the model quite
well, except for the aforementioned time in late August. (Westerlund and Tuomi, 2016) presented a similar comparison to a
different model configuration, derived from an earlier version of Nemo-Nordic. That model used different atmospheric forcing
fields taken from an operational HIRLAM forecast from the FMI (Finnish Meteorological institute), climatological boundary
conditions, climatological river runoffs and initial conditions from FMI's operational Baltic Sea forecast. Furthermore, it
did not have the light penetration parameterization present in the official Nemo-Nordic configuration described in this paper.
Compared to those results, the near-surface temperature in the official Nemo-Nordic results differs more from the observations
in autumn, but shows less bias in early summer. Thermocline depths were quite similar in both configurations.

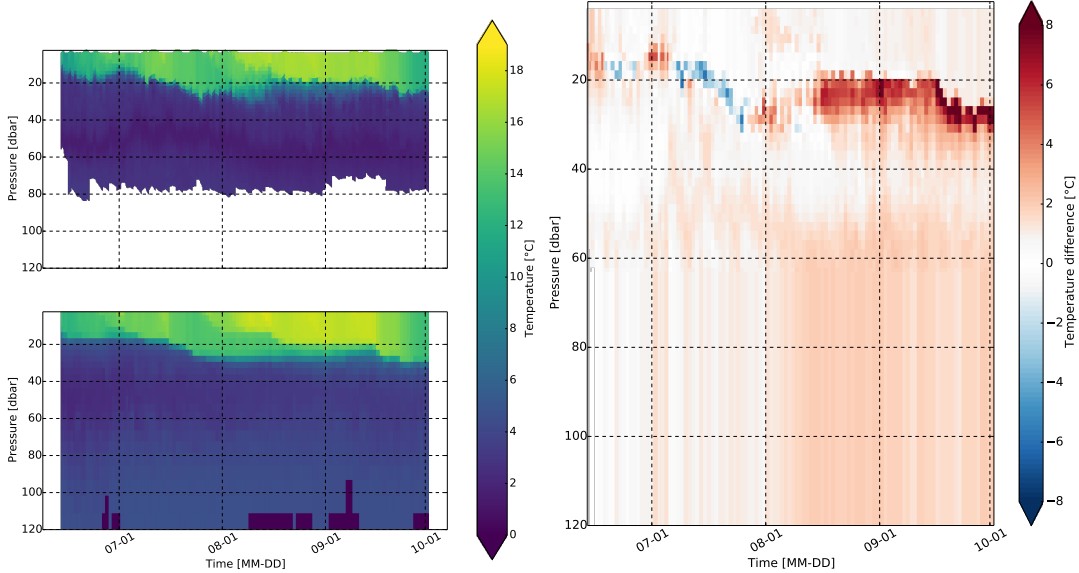

**Fig. 11.** Comparison of temperature profiles from an Argo float (upper left panel) and Nemo-Nordic. Observational data in the upper panel
has been redrawn from the dataset used by Westerlund and Tuomi (2016). The panel on the right hand side shows the difference between
the observation and model run. The model was larger than the measurement where the difference is positive. Model results have been taken
along the buoy route in the Bothnian Sea in 2013.

## 4.3 Thermohaline structure of the North Sea

Compared to the Baltic Sea, the North Sea is more homogeneous regarding its thermohaline structure. It exhibits mostly
seasonal variations in the form of the formation of the seasonal thermocline and seasonal variations in freshwater forcing.
Haline fronts between coastal and offshore areas are found in the Southern and Eastern North Sea due to the relatively large
river runoff from continental rivers at the Southern shore of the North Sea, and the Norwegian Coastal Current carrying low
saline waters from the Baltic Sea.

### 4.3.1 Vertical structure and seasonal variability

In this section we validate the vertical structure and seasonal variability at four stations representative for different hydrological
regimes in the North Sea. Two stations are located near the boundaries towards the Atlantic Ocean (Fladen Ground and the
Southern Bight). Validating the temperature and salinity structure in these areas also gives a validation of the boundary
conditions and the properties of the inflowing water. The open boundary condition fields for tracers (T and S) come from a
climatology in this case, so it is interesting to check whether their use with the model enables it to represent measurements.
The two other stations (NCC and Frisian Front), are located in areas where there are relatively large horizontal and vertical
(only NCC) salinity gradients due to the freshwater forcing from the Baltic Sea and the continental rivers draining into the
Southern North Sea. The observational data comes from the KLIWAS dataset provided by the University of Hamburg (Bersch
et al., 2013). It is composed of all available measurements between 1970 and 2013 in the North Sea, that have been put into a
1x1 grid.

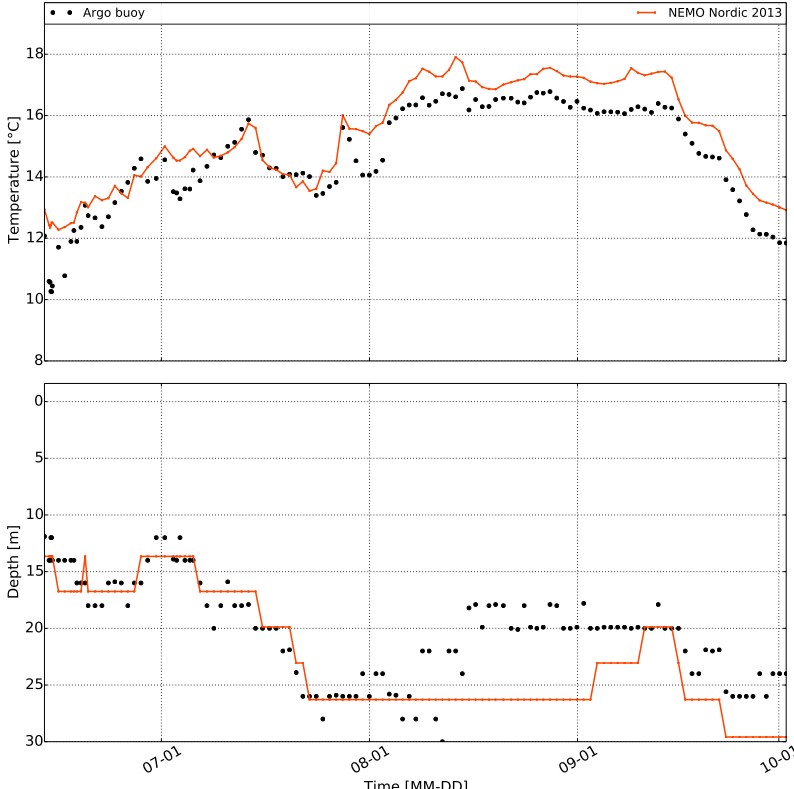

**Fig. 12.** Near-surface temperature in 2013 from the Argo float data and from Nemo-Nordic. Thermocline depth estimated as the maximum of vertical temperature gradient in the lower panel.

The haline vertical structure at the four stations in the North Sea is displayed in Figure 13. The only station with a distinct permanent haline stratification is the NCC station. The vertical haline structure at this station is well reproduced by the model, although the surface waters are less saline than the observations by about 1 PSU. The other stations are rather homogeneous in the vertical with respect to salinity. In the Southern Bight and the Fladen Ground the model captures the overall mean salinity. At the Frisian Front the modelled salinity is about 1.5 PSU too low, which probably is related to a displacement of the front between coastal waters with lower salinity and offshore waters. At all stations the model simulates a seasonal cycle in the surface salinity with a freshening during the summer months, in agreement with the observations. The timing and the amplitude of this summer freshening is however subject to some biases. Because the dataset does not contain regular measurements from these positions it can on the other hand give rise to biases in the observational estimates of the seasonal cycle.

The thermal vertical and seasonal structure is, as for the Baltic Sea, dominated by the seasonal warming and cooling of the surface waters (Fig. 14). In the Southern Bight and at the Frisian Front the waters are well mixed from surface to bottom throughout the year, and no seasonal thermocline develops, which is well reproduced by the model. In the Southern Bight there is a warm bias in the order of 0.5 °C (annual mean) in the model throughout the water column. At the Frisian front there is a warm bias of about 0.3 °C in the surface waters. At the two deeper stations, the seasonal development, and the depth, of the thermocline is well reproduced in the model, although the start of the thermocline formation is somewhat too early in the model. This is the case especially at the Fladen Ground, where it starts almost one month too early. Also the winter SST's are too warm in the model, resulting in a too weak winter convection/ too warm temperatures of the convecting water, which in its turn gives a warm bias in the deep waters. At both stations there is a warm bias of about 0.5 °C (annual mean) in the surface waters.

### 4.3.2 Modelled thermocline dynamics

For biogeochemical processes summer thermal conditions are important when water temperature and light intensity stimulate the growth of phytoplankton. Contemporaneously, thermal stratification develops in the deeper basin of the northern North Sea

## Salinity (psu)

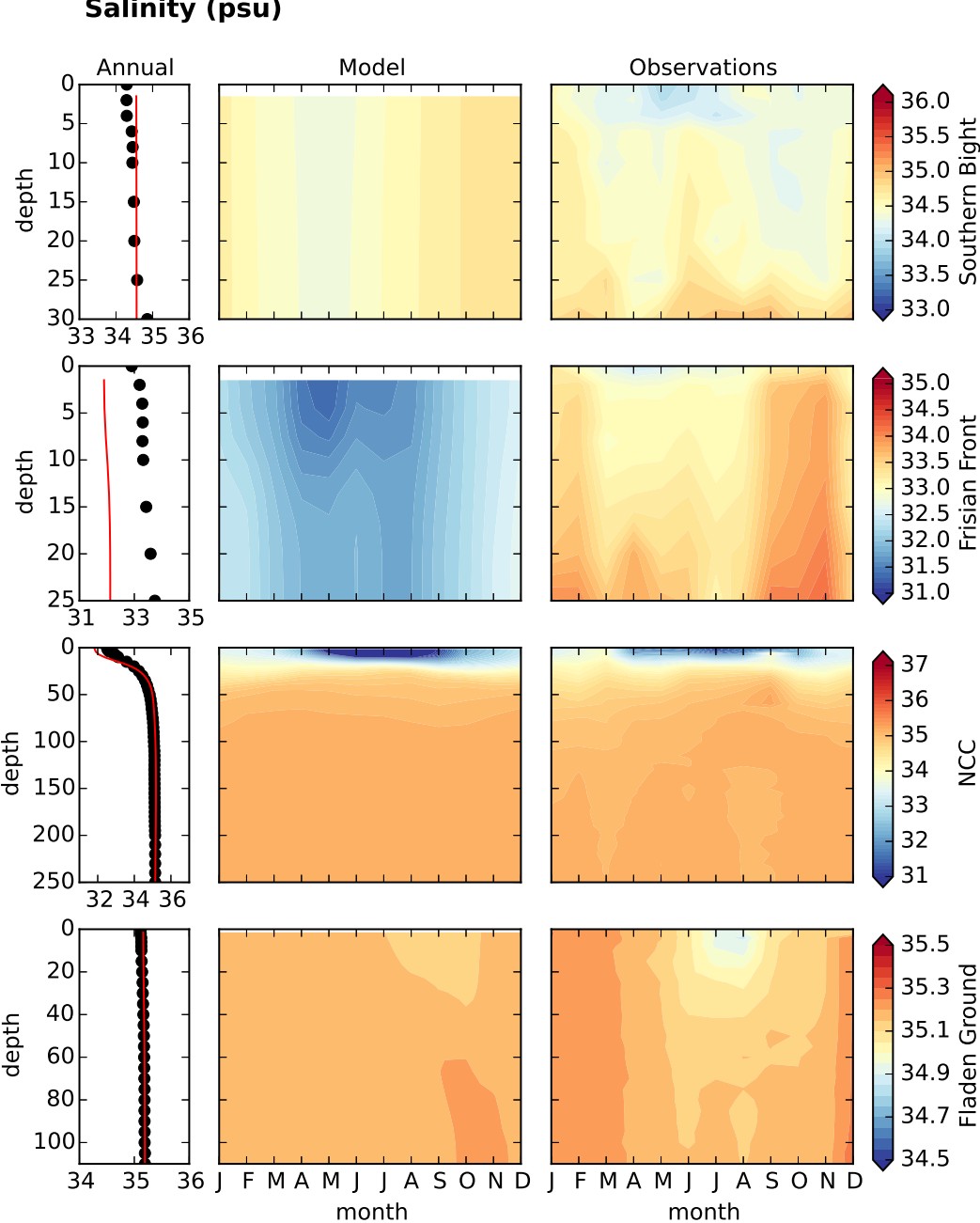

**Fig. 13.** Haline structure for four different stations of the North Sea, from top to bottom Southern Bight station, Frisian Front station, NCC station, and Fladen Ground station. The left column displays annual mean depth profiles, where the red line is simulated salinity and black dots are observations. The middle and the right columns show seasonal variations in the model and the observations, respectively. The observations are based on a 1979-2010 climatology.

# Temperature (deg. C)

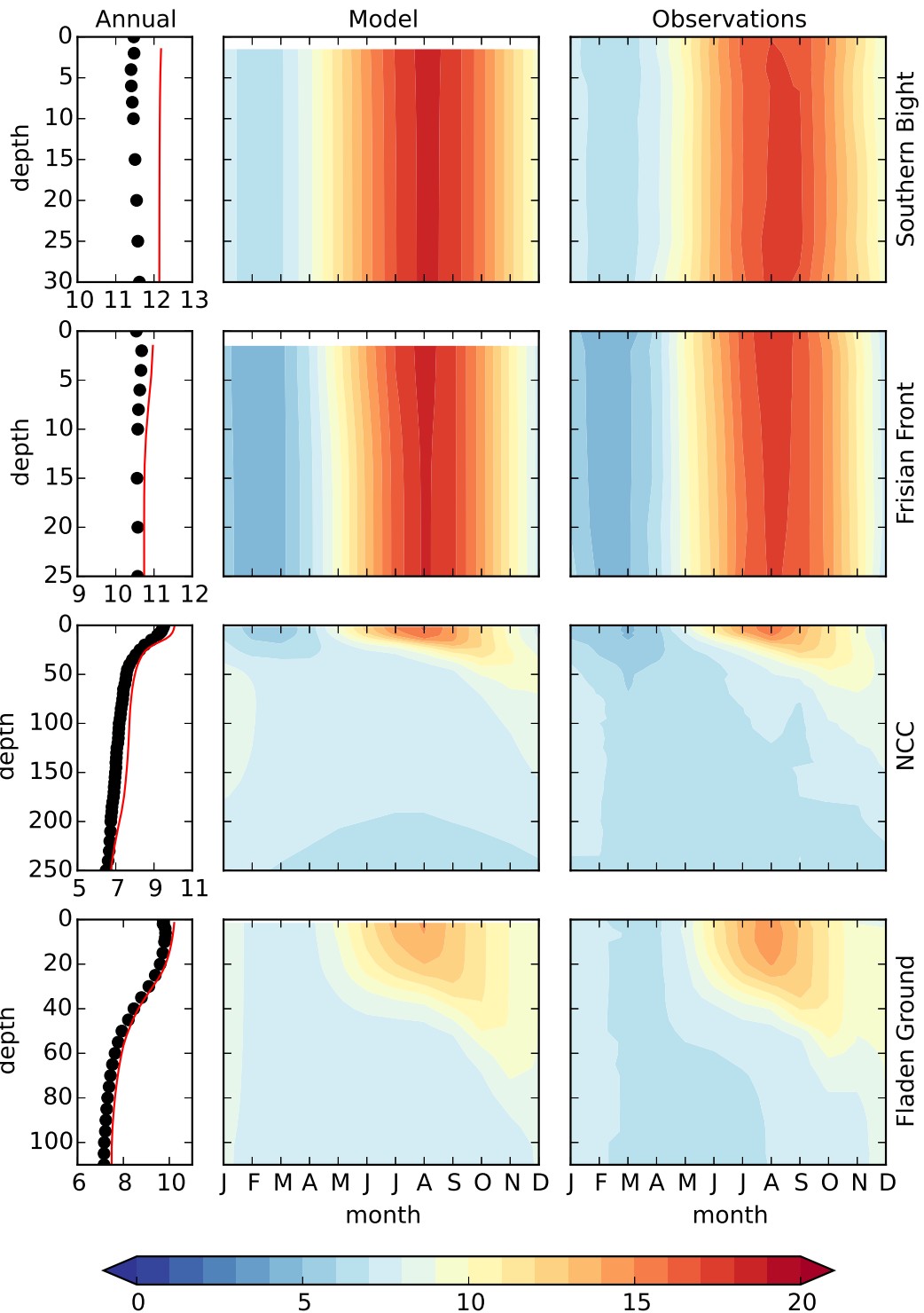

**Fig. 14.** Thermal structure for four different stations of the North Sea, from top to bottom Southern Bight station, Frisian Front station, NCC station, and Fladen Ground station. The left column displays annual mean depth profiles, where the red line is simulated temperature and black dots are observations. The middle and the right columns show seasonal variations in the model and the observations, respectively. The observations are based on a 1979-2010 climatology.

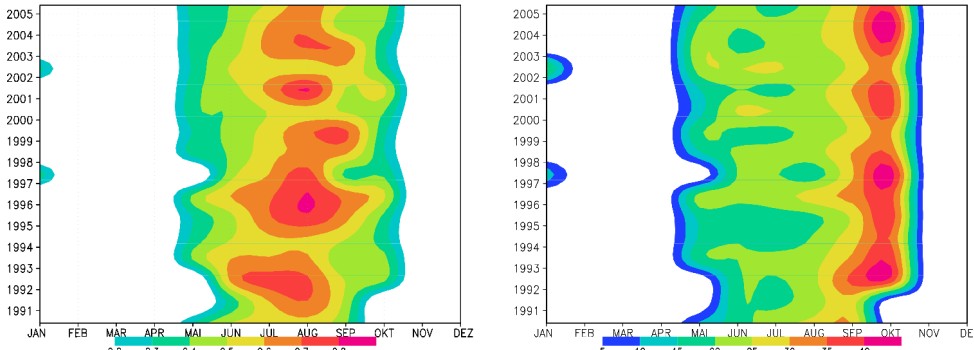

**Fig. 15.** Seasonal cycle of thermocline structure. Left: thermocline intensity (Degrees m$^{-1}$) averaged over the entire thermal stratified area. Right same as left but for for thermocline depth (m).

and the inflow of Atlantic water weakens (Skogen et al., 2011) and accordingly atmospheric forcing at the surface becomes more important. This leads to substantially better reproduction of interannual variability even in the North (Fig. 16b). We analyze in the following the modelled thermocline dynamics following previous approaches for the North Sea (Pohlmann, 1996; Meyer et al., 2011; Mathis and Pohlmann, 2014), which define the presence of thermocline conditions when a certain vertical temperature gradient is exceeded. We here chose a critical gradient of 0.25 K/m. The yearly maximum extent of stratified areas is primarily governed by topography and wind stress (Mathis and Pohlmann, 2014) and varies between 1990 and 2005 between 100,000 and 185,000 km$^2$. Stratified conditions begin to develop in May and reaches its maximal intensity during July and August. Already during September wind stress strengthens and temperatures lower which increases mixing. The thermocline weakens then and is shifted downward. As a result, nutrient rich water reaches the euphotic zone. During this time a second phytoplankton bloom can be sometimes observed in the North Sea (van Haren et al., 2003; Moll, 1998).

### 4.3.3 Interannual variability

Surface properties are in many cases dominated by local meteorological forcing (Skogen et al., 2011). Therefore, its important to validate also subsurface properties which are also influenced also the by circulation dynamics (like e.g. Atlantic inflow). For this, we use the KLIWAS observational data set (Bersch et al., 2013). We here follow previous approaches and subdivide the North Sea into 1° x 1° boxes for which for each month, at each box and standard level Taylor statistics (Taylor, 2001) is calculated (for details see Raddach and Moll, 2006, Gröger et al. 2013). In order to estimate the interannual variability of seasonal signals we chose January and July as representative for winter and summer as in these month observations were most abundant in the observational data set. The results are shown in table 5. In total 25619 observations were used for the analysis.

Overall good correlation with the temperature and salinity observations for summer and winter indicates the models skill to capture interannual variability and thus, the ability to realistically respond to low frequency modes of climate variations such as decadal variability or the NAO which likely influence the North Sea hydrography (e.g. Hjøllo et al.2009, Mathis et al., 2015). The root mean square values lie well within the standard deviation of the observations (with the exception of January salinities where rms is slightly higher,table 5). The models salinity variability (as indicated by standard deviation) is too high (∼50%), but the spatial distribution of the observations is much coarser than the model grid which explains this result.

### 4.3.4 Comparison with satellite data

In the following we compare the modelled SST with a widely used satellite product provided by the Federal Maritime and Hydrographic Agency of Germany, Hamburg (Bundesamt für Seeschifffahrt und Hydrographie, BSH hereafter). By this we are able to investigate how the modelled SST reproduces interannual variability which is important for the application of the model for climate services and the simulation of various climate scenarios.

In contrast to the warm bias relative to the in-situ temperature measurements, the simulated winter SST (Fig.16) is almost everywhere colder than the satellite estimations of SST. Cold biases relative to satellite measurements were also obtained in

**Table 5.** Quantitative comparison of simulated and observed state variables derived from Taylor statistics (Taylor, 2001). rms = root mean squared, corr = Pearson's correlation, stddev = standard deviation, N = number of observations. The statistics have been derived from all 1° x 1° boxes. See text for details about data set and data handling.

| State variable | rms | corr | stddev observation | stddev simulation | N |
|---|---|---|---|---|---|
| | | | Winter | | |
| Temperature | 0.74 | 0.78 | 0.86 | 1.18 | 6139 |
| Salinity | 0.56 | 0.79 | 0.54 | 0.88 | 6034 |
| | | | Summer | | |
| Temperature | 1.10 | 0.94 | 3.06 | 3.15 | 6808 |
| Salinity | 0.56 | 0.83 | 0.72 | 0.98 | 6638 |

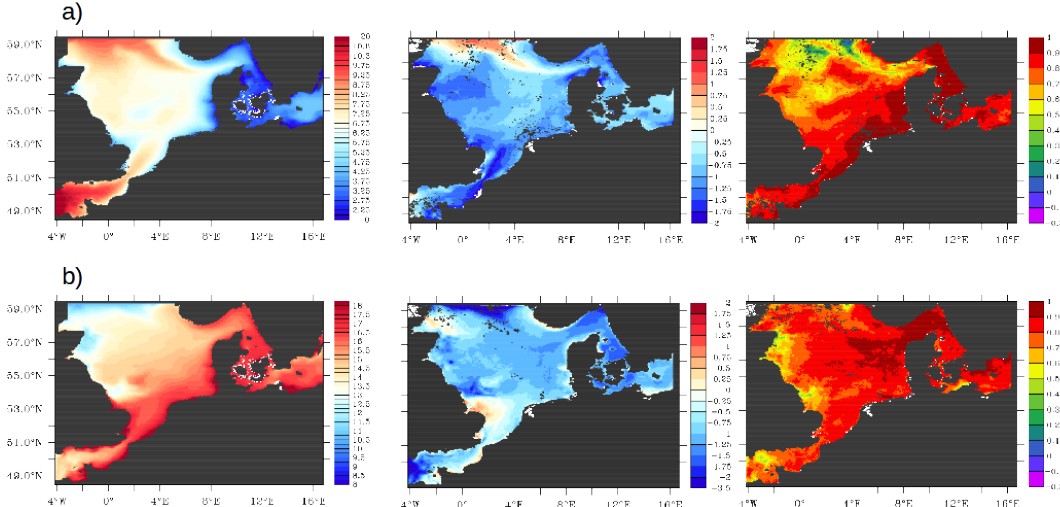

**Fig. 16.** a) left: Nemo-Nordic multiyear (1990-2005) DJF average sst. middle: Nemo-Nordic minus BSH sst. right: Correlation between Nemo-Nordic sst and BSH sst. Note regions where not enough observational data were present have been coloured grey. b) same as a) but for JJA.

other ocean models driven by the ERA40 reanalysis dataset in (Tian et al., 2013); (Gröger et al., 2015)). Given that in-situ temperature measurements give a more direct measure than satellite estimations of SST's, and that the atmospheric forcing used for these simulations has a warm bias, especially in the North Sea during winter (Landelius et al., 2016), it is likely that the satellite underestimates the SST. Despite this, the satellite SSTs can be used for validation of spatial and temporal variability.

Figure 16 (a) shows an overall good correlation of modelled interannual SST variations with the BSH data set. Likewise the lower correlation in the Northern deeper parts could indicate problems with convective and/or wind induced mixing which would influence the heat transfer from the deeper layers to the surface. In contrast, in the Southern and shallower parts, SSTs are dominated largely by the atmospheric forcing (Skogen et al., 2011) and, thus correlation increases. Good correlation is also seen along the Norwegian coast where the effective heat capacity is limited due to haline stratification which make SSTs more sensitive to the atmospheric forcing. The same is true for nearly the entire Baltic Sea where correlation almost nowhere drops below 0.8.

## 5 Conclusions

In this article we provide a detailed description of the dynamic features of Nemo-Nordic, a newly developed joint setup for the Baltic Sea and the North Sea. Its performance when it comes to sea-ice is the subject of a specific article on its own (Pemberton et al., 2017). We have shown that Nemo-Nordic is able to reproduce the barotropic and baroclinic dynamics, as well as the thermohaline structure, of Baltic & North Sea basins. The key to achieve this overall good representation of the physics in the Baltic Sea and the North Sea has been to get a representation of the barotropic dynamics as good as possible. This ability, which is detailed in the present article, has been validated with the most demanding procedure: Nemo-Nordic is used as the official forecast model of SMHI for Baltic & North Seas, including for SSH for which the model does not benefit of any data assimilation. In forecast mode, Nemo-Nordic is used with a higher resolution (1nm or 1852 m), compared to longer integrations where the setup is used with a 2nm (or 3704 m) resolution. The ability of the model to represent barotropic dynamics at high frequency time scales (hours to several days) is the main reason for its good representation of transports and water exchanges in and between the Baltic Sea and the North Sea. It also allows for an accurate representation of the Baltic Sea inflows, and the initiation of the Baltic Sea baroclinic dynamics. In order for Nemo-Nordic to proper represent baroclinic dynamics in both basins, a specific tuning of mixing is done from both horizontal and vertical points of view. For example a limitation of the vertical mixing length in order to help the model keep the Baltic Sea halocline at a realistic level.

Although Nemo-Nordic well reproduces the overall physics of Baltic & North Sea, some biases can be noticed. The sea level representation is better in forecast mode than that of the previous SMHI operational model HIROMB, but some improvements are still needed to better represent extreme events. Regarding this precise point, a coupling with wind waves appears to be the next step, as it was recently shown that the contribution of wind waves has a major impact on extreme sea levels (Staneva et al., 2016). Ongoing development based on the new implementations within the NEMO ocean engine concerns the inclusion of wetting and drying processes which can also affect sea level in shallow areas. Further, an important margin of improvements exist in atmospheric forcing, bottom friction and bathymetry. A margin that should lead to an even greater accuracy in sea level representation: recent tests done with a bathymetry computed from the GEBCO database (The GEBCO-2014 Grid, version 20150318, http://www.gebco.net) show a high improvement of sea level representation in the North Sea and along the Swedish West coast.

From a baroclinic perspective, several aspects can be improved in the model. First, even though we could show that the barotropic variability of the Baltic & North Sea basins allows the Baltic Sea MBIs, there is still a bias in the deep salinity due to an over ventilation of the intermediate layers. So far, the only solutions which were found to solve this bias is to increase resolution. But a major future development would be the use of a hybrid vertical coordinate system with $z^*$ close to the surface and $\sigma$ coordinates close to the bottom. A minor development concerns the Galperin parameterization as mentioned before, which needs to be decoupled from thermal dynamics. Ongoing tests are being made concerning this latest point. The geometry of the horizontal diffusivity is to be improved as well in order to limit its effect on Baltic Sea inflows. The underlying idea is to avoid any horizontal diffusivity close to the bottom. Other developments in Nemo-Nordic concern its coupling with biogeochemical models. Nemo-Nordic has been coupled with the SCOBI model (Eilola et al., 2009), for which a validation work is ongoing. Nemo-Nordic has also been coupled with the BFM model (Vichi et al., 2007, 2015) on a part of its domain (Fransner et al., 2017).

Nemo-Nordic, and spin-off configurations, provide a tool not only for ocean forecasting, but also for a wide variety of ocean research. It can be used for long term simulations either for process purpose studies (Hordoir et al., 2013; Godhe et al., 2013; Moksnes et al., 2014; Westerlund and Tuomi, 2016, e.g.) or climate change related studies (Hordoir et al., 2015; Höglund et al., 2017). It can be also used for biogeochemical-ecosystem studies, using a simple passive tracer with a decay rate (Fransner et al., 2016) or coupled with a complex biogeochemical model (Fransner et al., 2017). Nemo-Nordic is also the ocean component of an RCA4-NEMO coupled model which is the basis of a regional climate model used in several studies (Wang et al., 2015; Pätsch et al., 2017).

Finally, Nemo-Nordic has also been used as a boundary condition for high-resolution sub-basin scale setups (Westerlund et al., 2017).

## 6 Code and data availability

Nemo-Nordic builds on the standard NEMO code (`nemo_v3_6_STABLE`, revision 5628) with only minor changes, including the fast-ice parametrization and a spatial varying background viscosity/diffusivity that could be read in from the file. The standard NEMO code can be downloaded from the NEMO web site (http://www.nemo-ocean. eu/). The `nemo_v3_6_STABLE`

version is available from the following link: http://forge.ipsl.jussieu.fr/nemo/vn/branches

590 /2015/nemo_v3_6_STABLE. The new code blocks that are introduced (relative to the standard NEMO code `nemo_v3_6_STABLE`, revision 5628) into our Nemo-Nordic code are included as supplemental material. Nemo-Nordic is released under the terms of the CeCill license (www.cecill.info) and its code is available in the zenodo archive. Access to the forcing data, analysis scripts, and data used to produce the figures in this study can be made available upon request to the corresponding author.

## 7   Author Contribution

Most of the development and design of this model was conducted by Robinson Hordoir. Lars Axell helped tuning several parameterers especially within the turbulence scheme. Anders Höglund and Christian Dieterich provided expertise and help to design forcing and boundary conditions. Filippa Fransner conducted several experiments and designed diagnostics. Matthias

Gröger, Ye Liu and Per Pemberton provided ocean modelling expertise. Semjon Schimanke provided an important expertise regarding atmospheric forcing. Helen Andersson provided scientific expertise and management. Patrik Ljungemyr, Petter Nygren, Saeed Falahat, Adam Nord, Anette Jönsson, Iréne Lake provided expertise in the design of the operational version of Nemo-Nordic. Kristofer Döös, Magnus Hieronymus, Heiner Dietze, Ulrike Löptien and Ivan Kuznetsov provided expertise and advice on the representation of processes for the Baltic and North Sea area. Antti Westerlund, Laura Tuomi and Jari Haapala

provided comparisons with measurements and expertise on sea-ice modelling. All the authors contributed with comments to the manuscript.

*Acknowledgements.* The research presented in this study is part of the project BONUS STORMWINDS and has received funding from BONUS, the joint Baltic Sea research and development programme (Art 185), funded jointly from the European Union's Seventh Programme for research, technological development and demonstration and from the Swedish research council for environment, agriculture sciences and

610 spatial planning (FORMAS). This work has been also supported by the Strategic Research Council at the Academy of Finland, project SmartSea (grant number 292 985). The Nemo-Nordic simulations were conducted on the Bi supercomputer of the National Supercomputing Center, Linköping University, Sweden. The corresponding author wants to express his gratitude to Sofia Bergenbrant, jurist at SMHI, for the energy she provided to encourage the Nemo-Nordic technology to be exported outside SMHI.

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
