# Peer review of "Nemo Nordie 1.0: A NEMO Based Ocean Model for Baltic & North Seas, Research and Operational Applications"

_Geoscientific Model Development, 2018_

## Referee Comment (RC1) · Anonymous Referee #1 · 20 Apr 2018

The authors present a modeling study of the North Sea and the Baltic Sea with the NEMO model. The strong point of this modeling is the representation in the same high-resolution domain of these two seas and their exchanges through the Danish strait. The paper presents a number of comparison with observations validating the model. I believe this paper will be of interest to the community of NEMO users and more generally to researchers interested in the North and Baltic Seas. It can therefore be published with minor revisions. I propose (see following) some remarks that I leave to the appreciation of the authors to improve certain aspects of their article.

lines 40-55: a figure with a map of the main currents and key elements of local dynamics could be welcome

line 55-65: the authors emphasize the importance of the transition between the two seas (North and Baltic). Is the horizontal resolution sufficient to represent all these small straits? A mesh of almost 4km seems a bit coarse to achieve this goal.

line 67: if possible I would suggest replacing the nauticmiles by the metric system

line 96: I think we could be more specific: for example explain what are the physical processes represented by NEMO that allow NEMO to be better adapted than these other models cited in lines 95.96

line 96: " the dense overflows that feed its very specific sill bounded estuarine circulation ". Unclear. . .. Detail a little more.

line 101: the authors specify that they use version 3.6 of NEMO. If this seems relevant to the authors, I would suggest to say what this version brings compared to previous versions.

line 105: This area has a large overlap with the NEMO-IBI operational system domain (Maraldi et al, 2013). The horizontal resolution is ultimately worse than that of IBI but NEMO-NORDIC brings the connection with the Baltic, missing from IBI. The overlapping zone also offers interesting intercomparison possibilities between the two models. The authors could say a few words about the respective interests and the complementarity of NEMO-NORDIC and NEMO-IBI and if possible make a reference to the validation study of NEMO IBI by Maraldi et al, 2013.

line 108: NEMO's recent advances on the sigma coordinate could be cited here. I think for example to the paper of Shapiro 2013 (or other if the authors see a reference more relevant)

line 112 "adopted" "adapted"

L132: the word "decouples" seems clumsy to the extent that there is in fact a coupling

between baroclinic and barotropic modes.

L135: the term "degree of conservation" implies that the conservation properties of tracers may not be strictly respected, which seems a priori not very compatible with the study of climate. If the model really respects the conservation properties of tracers the authors should say it more clearly (or can refrain from commenting on a fairly basic property).

L140: Is the roughness given by a length of roughness? Is it a constant or a mapped parameter? Since this parameter seems important it would be interesting to give its value, or its order of magnitude if it is not a constant.

L145-150: without doubting the proper functioning of the BBL, is it still not unsatisfactory to have to completely remove the advective component of this parameterization? Independently of the numerical considerations that one has well understood, is it not less realistic from the point of view of physics? Would that not finally plead for the use of the sigma coordinate referred to online 107?

lines 150-160 What is "tuning" mentioned by the authors is not very clear. Do the authors refer to the value of the Galperin coefficient? In this respect, the reference to Galperin's paper is too vague. One could for example think that one refers to the functions of stability of Galperin? Is that the case? According to Reffray et al 2015 the choice of Canuto seems the most judicious... Or do you refer to the limitation of the mixing length (eq22 of Galperin)? What exactly is this coef mentioned above? The problem raised by the authors also refers to the choice of the thresholds of minimum values for the TKE of the closure scheme and their possible regionalization. Can the authors say a few words about the values used by NEMO?

lines 165-175 The authors mention the drawbacks of the calculation of the horizontal mixing in z coordinate, which, even taking into account the NEMO rotation tensor, tends to introduce a significant diapycnal mixing. The authors correct this defect by means of a spatial adaptation of the coefficients of viscosity / diffusivity which seems a little

artificial but which has the merit of working. There is certainly room for discussion of possible future improvement prospects for NEMO-NORDIC. Insofar as the saline intrusions evoked by the authors would follow the bottom, one can for example wonder if the sigma coordinate would not be better adapted. The work of Shapiro 2013 on the different forms that the sigma coordinates can take in NEMO could be a source of inspiration for a possible evolution of NEMO-NORDIC in this direction.

line 183: the tide is apparently introduced as a boundary condition only. Does this mean that the internal generating forces in the numerical domain (astronomical potential and loading self attraction present in the NEMO version used by Kodaira et al, 2016) are not used here? If yes, why? Are they negligible in comparison with the influence of boundary conditions?

lines 184 The open boundary conditions seem to have a fairly high level of elaboration with respect to the barotropic processes (tide, storm surge). On the other hand, I am surprised, given the possible operational purpose, and also given the Copernicus context in which the NEMO-NORDIC model seems to be developed, by the great simplicity of the boundary conditions for the general 3D circulation. Only temperature and salinity seem to be concerned (nothing specific is said about SSH and currents, apparently). In addition, T and S would be climatological. This simplicity can be understood in the context of a climate projection, but for operational applications it is expected that the Coperninus operational system will serve to provide boundary conditions for regional models such as NEMO-NORDIC. I may have misunderstood the text which in this case should be a little clarified. Note also that the IBI operating system seems to have the capacity to forecast storm surges according to Maraldi et al 2013. Can the authors discuss a little more about their choices?

line 185. The authors apparently use the TPXO tide atlas of Egbert et al 1994. In Maraldi 2013, the accuracy of altas FES is widely commented and finally used as a reference to validate the quality of the tide simulation obtained with NEMO. The present study could have been an opportunity to make a comparison between the different tidal

atlases usable as boundary conditions. Which produces the best result? etc etc... That would be useful I think. This is a minor remark but if the authors deem it appropriate they might mention this fact as a possible prospect in future work.

line 195-205 The introduction and the abstract of the article suggest that NEMO-NORDIC is used for both climate studies and short-term operational forecasting. The description of the atmospheric forcing seems to correspond to the first point only. What is the authors' strategy for short-term operational forecasting? Does the hourly forecast of the sea level for example impose particular constraints with regard to the frequency of the atmospheric forcing? We can also think that the precise prediction of the sea level requires taking into account the effect of the waves. Preliminary developments have been made in NEMO on this subject (see, for example, NEMO and WW3 coupling by Clementi et al, 2017 for a better representation of the drag coefficient). What is the authors' strategy for this question?

lines 208-215 About taking into account a constant concentration of chlorophyll to improve the essential point of the penetration of light. Would there be an interest (perspective) in using Copernicus' global predictions of chlorophyll?

line 243: I am surprised when the authors say that the correlation is mostly close to 0.99. I would have rather said 0.95. This difference of appreciation is probably subjective and attributable to the lack of readability of Figures 2-3-4. In fact it seems to me that Taylor diagrams are not very suitable here. Figures 2-3-4 indeed occupy a lot of space for little information (only two points, a yellow, a blue) with a very low level of readability since each individual figure per tide station is finally tiny. We therefore lose a lot of time trying to see what are the RMS values, Standard deviation, correlation, when a simple table would immediately give this information, and allow a quick comparison with other authors (see for example Table 1 in Maraldi 2013). It seems to me that Taylor diagrams are appropriate when a single reference is compared to a scatter plot. For example, in Toublanc et al (2018) Figure 7, it is immediately understood that the simulations corresponding to green and blue point clouds are better than the simulation

giving the cloud of red dots.

Note in passing that the altimetry is a powerful tool for validation of the tide in the regional or global models, not subject to the possible strong local specificities (harbor installation, particular bathymetry) which characterize eventually the coastal gauges and that a 4km resolution model can not represent (see Toublanc et al, 2018, kodaira et al, 2016).

Finally it would not be useless to present a map of the amplitude and the phase of the main wave M2 (and possibly the M4 wave which allows to appreciate the accuracy of the effects of non-linearity M2-M2 in a model) to allow quick comparisons with previous studies (NEMO in Maraldi 2013, T-UGOm 2D in Pairaud et al 2008, etc, etc ...).

line 247: The issue of the horizontal resolution is appropriately addressed in the Strait of Denmark. Some passages are indeed so narrow that a resolution of 2nm (almost 4km) seems clearly insufficient. For example the passage between Elsinore and Helsingborg barely fits a mesh. However, in NEMO, there is possibility for local increase in horizontal resolution, either by using an AGRIF nesting (Waldmann et al, 2016), or by using the NEMO curvilinear grid (Madec Imbard, 1996). To what extent can either of these two possibilities constitute a NEMO-NORDIC development perspective? The thresholding of the bathy (note that Maraldi 2013 also uses a threshold and discusses its consequences) also seems problematic: is not it a handicap for the forecast of surges? What is the technical reason that prevents lower bathymetry? Could the sigma coordinate overcome this problem?

Table 1: Northern boundary, the "Inflow Observations" bounds are given in descending order. Is it correct?

line 328. My next question is motivated by the authors' commentary on the Galperin coefficient and the fact that NEMO has two types of turbulent closure (TKE, k-epsilon, Reffray 2015). In the manner of Reffray et al 2005, have the authors made a sensitivity test of the SST bias to the turbulent closure scheme (TKE or K-epsilon)?

[Figure]

lines 353-355: In the description of the model it would therefore be interesting to say a few words about the state equation used.

line 370. It seems to me that the mechanisms responsible for the Major Baltic Inflows of 1993 and 2003 could be explained in more detail if possible (would there be no more things to say outside of the sea level? ?). Do these mechanisms come from open boundary conditions, or are they generated inside the modeling domain etc etc ...?

line 393. Specify the number of the Figure

line 415: the authors evoke the possibility of a validation of their boundary conditions. But what exactly are the boundary conditions for T and S? How are the T and S fields constructed that force the open-border model? Climatology, ORCA25? Same remark for baroclinic currents.

Figure 14: Would a single color palette (as in Figure 15) be preferable?

line 481: it seems to me that the acronym BSH is not defined.

References:

Clementi et al, Coupling hydrodynamic and wave models: first step and sensitivity experiments in the Mediterranean Sea. Ocean Dynamics (2017) 67:1293–1312 DOI 10.1007/s10236-017-1087-7

Kodaira, T., K. R. Thompson, and n. p. Bernier (2016), Prediction of M2 tidal surface currents by a global baroclinic ocean model and evaluation using observed drifter trajectories, J. Geophys. Res. Oceans, 121, 6159–6183, doi:10.1002/2015JC011549.

Madec et al 1996: A global ocean mesh to overcome the North Pole singularity Climate Dynamics May 1996, Volume 12, Issue 6, pp 381–388

Maraldi C., Chanut J., Levier B., Ayoub N., De Mey P., Reffray G., Lyard F., Cailleau S., Drevillon M., Fanjul E. A., Sotillo M. G., Marsaleix P., 2013. NEMO on the shelf: assessment of the Iberia–Biscay–Ireland configuration. Ocean Science, 9, 745–771.

http://dx.doi.org/10.5194/os-9-745-2013

Pairaud I. L., Lyard F., Auclair F., Letellier T., Marsaleix P., 2008, Dynamics of the semi-diurnal and quarter-diurnal internal tides in the Bay of Biscay. Part 1: Barotropic tides, Continental Shelf Research, 28, 1294-1315 http://dx.doi.org/10.1016/j.csr.2008.03.004

G. Reffray, R. Bourdalle-Badie, and C. Calone, Geosci. Model Dev., 8, 69–86, 2015 www.geosci-model-dev.net/8/69/2015/ doi:10.5194/gmd-8-69-2015

Shapiro et al, 2013: Ocean Sci., 9, 377–390, 2013 www.ocean-sci.net/9/377/2013/ doi:10.5194/os-9-377-2013

F. Toublanc, N.K. Ayoub, F. Lyard, P. Marsaleix, D.J. Allain, Tidal downscaling from the open ocean to the coast: a new approach applied to the Bay of Biscay, Ocean Modelling, Volume 124, April 2018, Pages 16-32, ISSN 1463-5003, https://doi.org/10.1016/j.ocemod.2018.02.001.

Waldman, R., Herrmann, M., Somot, S., Arsouze, T., Benshila, R., Bosse, A., . . . Testor, P. (2017). Impact of the mesoscale dynamics on ocean deep convection: The 2012–2013 case study in the Northwestern Mediterranean Sea. Journal of Geophysical Research: Oceans, 122. https://doi.org/10.1002/2016JC012587

---

## Referee Comment (RC2) · Anonymous Referee #2 · 29 May 2018

**1 Overview**

This system description and evaluation paper constitutes a useful contribution to the ocean modelling community at large and in particular to researchers focused on the North and Baltic Seas.  The authors make a clear case for the potential advantages from modelling both basins and their interconnection all in one system. They also note some of the difficulties that entails and some of the compromises that are required to model these two basins with considerably different dynamics.  The paper is generally well written. There are a surprising number of typos and technical inconsistencies but nothing a copy editor should not pick up on.  There were a few instances where the

text was not particularly clear but overall it is of a good quality. The paper gives a useful assessment of the model skill compared to relevant observations in each basin on a variety of time scales. I believe the paper with some minor corrections should be put forward for publication. Below I have some comments that the author may wish to consider to help refine the paper a further.

2 General Comments

C1 I think within the introduction (∼L85) or the Model set up needs mentioning of the two differing resolution models that are referred to later in the paper. It is not immediately obvious which version of Nemo-Nordic is being assessed at any one time, especially as both are later compared against each other. I think it would help the reader if there was some way to make this clearer, e.g. Nemo-Nordic 1nm/2nm etc. or some other similar labelling strategy early in the paper and a description of these. In the model description there is only a description of the 2nm version. Perhaps restricting nautical miles to metric equivalents will be more in line with GMD.

C2 L105. With regards to the 2nm grid description, it might be useful to state if the grid is rotated, otherwise it would be hard to see how the stated grid resolution would be retained at a relatively high latitude.

C3 L113-L116 The stated vertical resolution is surprisingly coarse in a regional model. I appreciate there is a need to focus resolution with regards to the overflows but 3 m surface resolution seems quite low. I refer the authors to Stewart et al. with regards to what would be an optimal vertical resolution for a z-level model in a global context. K.D. Stewart, A.McC. Hogg, S.M. Griffies, A.P. Heerdegen, M.L. Ward, P. Spence, M.H. England,Vertical resolution of baroclinic modes in global ocean models,Ocean Modelling,Volume 113, 2017,Pages 50-65,https://doi.org/10.1016/j.ocemod.2017.03.012. Towards the end of the paper there is an analysis compared to an SST product. The bias is surprisingly large and cold given the warm bias in the atmospheric forcing, could the surface resolution play a part? What is defined as SST in this context?

[Figure]

C4 L124 The issue of model resolution and the Danish straits is correctly brought to the attention of the reader and the method by which the barotropic flux can be maintained by retaining the same cross-sectional area. However, this must be problematic with regards to the baroclinic part of the flow. Particularly so as one of the main motivations of having the interconnect is to model bottom saline intrusions form the North Sea that enter the Baltic. Perhaps there is justification here for some more comment on the effects on the baroclinic flows by attempting to retain the barotropic flux.

C5 L136 It is mentioned that 'tuning' is done with regards to optimizing model SSH. It is not clear what the optimization is, perhaps this could be elaborated as it could potentially save others time in the future or suggest useful strategies. I wonder could the authors supply a graphic/map in the supplementary material with regards to the 2d varying bottom friction "following the barotropic Kelvin wave", what is the physical grounds for this?

C6 L174-L175 The use of variable diffusivity and viscosity appears to be an interesting pragmatic engineering solution to the model difficulties concerning mixing and the dense water overflows. That is an interesting solution and appropriate for short time scale like a forecast model, but I wonder if it is appropriate for climate scales? That is this strategy assumes a-priori what the structure of the water column is, but on climate time scale that could change but the model may in effect be imposing it as it is, could the authors comment on this. It seems that as the authors note, a hybrid z* with sigma at the bottom is a much better vertical framework for the problem at hand. Could the authors comment on why such a huge viscosity is required at the boundary region? This is likely to cause severe issues for any coupled biogeochemistry model here. I suggest it is worth investigating what is happening to vertical velocity and tke here.

C7 L210 There are a number of chl products that are available. Do the authors consider using say even just a monthly climatology rather than a uniform value domain wise.

C8 Fig 2,3,4

I think the use of Taylor plots here is not appropriate, as there are only two data points. It could save a lot of space to reduce the Taylor plots to numerical tables. Taylor plots are beneficial when analysing a large 'cloud' of data. In a model sensitivity, they are useful if tuning say one parameter a number of times. In this case, there are just 2 model resolutions, it might be more appropriate if there are several model resolutions to intercompare. Enabling the modeller to visualise if there are say competing trends between rms, correlation and standard deviation. However, with just two data points there can be no trend to discern. In conclusion, a table might be quicker to interpret and save considerable space I do not think the Taylor plots here bring any advantage. With regard to the tides in the North Sea, the inclusion of a Co-tidal amp/phase plot of say M2 could be useful to give a quick look at how overall the model is doing in space with regards to tides.

C9 Fig 5

The climatological currents from North to South along the boundary in the English channel are very odd and suggest some problem in the bdy implementation here. Could the problem be related to using clim. TS in a highly tidal area? Ignoring the general cold bias in Fig 17, there is still an obvious bdy issue both in the north and in the south in JJA, again perhaps relates to the bdys provided. Too much vertical mixing??

C 10 L256 It is noted that the north sea underestimate lower frequencies but these are unbiased in the Baltic. (due to amplification?) Is there a case to be made that the model Is overdoing amplification of waves that are initially underestimated in the North Sea? If so could that have other adverse effects?

C 11 L 425 the authors show a large freshwater bias at the Frisian front location. May I also suggest that the riverine input from HYPE could be a possible issue here, Have the authors made an assessment of the HYPE model along this coastline? The accuracy is assessed for the Baltic Basin but not for the North Sea?

[Figure]

3 Minor Technical points/errata These are likely but a subsection of minor points that need further editing.

T1 The text in general refers to the configuration as Nemo-Nordic, but a number of the figures capitalise the term NEMO.

T2 L96, User of However needs a following comma, this occurs in a number of places.

T3 L108 "which is more simple as->than some Baltic Sea Models"

T4 L111 "showed-> shown"

T5 gaps between numbers and units often missing e.g. line114 60m instead of 60 m.

T6 L124. "The cross section(al) area"

T7 L229 "Benefits from the use(s)" T8 L239 "Straights->Straits"

T9 L252 "has a low->negative bias"

T10 L264 'which are one of the main driver(S) of the Baltic Sea'

T11 L265 : reword "The SSH in the Danish Straits bearing a high variability part along the coasts was less of a concern"

T12 Fig 2, change "up to bottom " to "top to bottom" in captions

T13 Fig 4 caption "North Sea and *English* Channel"

T14 Use of the term British Islands might want to be replaced with British Isles, e.g. L41

T15 L 329 In order to avoid these? Effects(s?)

T16 L346 "The Surface water at these station(s)"

T17 L369 "Major Baltic inflows *of* 1993 and 2003

T18 L340 "recorded duration(s)"
T19 L373 "this further suggest(s)"

T20 L383 suggest adding degree sign and N/E etc.

T21 L393 "are show(n) in the Figure *?*

T22 L 407 "It exhibits mostly seasonal variations in *the* form of .."

T23 Fig 16 caption,, seems "a)" is not required (Fig 17 ok) and depth [m] needs reformatting

T24 L456 "an subdivide" -> "and subdivide"

T25 L476 "data set in the" -> "data set in"

T26 L501 "proper(ly)"

―――――――――――――――――――――

---

## Author Comment (AC1) · 29 Oct 2018

**Revised Version Attached**

First we want to thank you for the constructive comments and the general work done on our manuscript. We answer each comment below. You ask a lot of question and show a great interest for our model and manuscript, which we appreciate very much of course. But sometimes we can not do all the work which you suggest, either for time related questions or because some questions (like for what generates the inflows for example) are still a subject of debate.

[Figure]

lines 40-55: a figure with a map of the main currents and key elements of local dynamics could be welcome

Indeed. We will add them. However, since such maps have been produced numerous times in many articles, we will just reproduce pre-existing one and cite their origin accordingly.

line 55-65: the authors emphasize the importance of the transition between the two seas (North and Baltic). Is the horizontal resolution sufficient to represent all these small straits? A mesh of almost 4km seems a bit coarse to achieve this goal.

It is too coarse to make any detailed study of the Danish straits, but an appropriate tuning can reproduce the ÂńÂăimpedanceÂăÂż of the Danish straits so that the transfers of volume, salt and heat between the two basins are represented. We mentioned this briefly in the text, but we will make it more clear.

line 67: if possible I would suggest replacing the nauticmiles by the metric system

We will put both.

line 96: I think we could be more specific: for example explain what are the physical processes represented by NEMO that allow NEMO to be better adapted than these other models cited in lines 95.96

We will detailed this part to a larger extent. Beyond the features present today in NEMO that other non-community models will likely never have, it's also about the yet non-exploited possibilities like wave coupling for example, or the future ones.

line 96: " the dense overflows that feed its very specific sill bounded estuarine circulation ". Unclear. . .. Detail a little more.

Done.

line 101: the authors specify that they use version 3.6 of NEMO. If this seems relevant to the authors, I would suggest to say what this version brings compared to previous

versions.

The biggest improvement was the new coupling between barotropic/baroclinic modes which enables a much better SSH representation, and since SSH variability is the driver of the entire system a better representation of almost everything. XIOS, and the ability to have only one output file and not one per processor anymore, made the model a lot nicer to handle for everyone. This is not a scientific argument but having motivated scientists to work on a model does help a lot. We will add more details about the scientific aspect in the text.

line 105: This area has a large overlap with the NEMO-IBI operational system domain (Maraldi et al, 2013). The horizontal resolution is ultimately worse than that of IBI but NEMO-NORDIC brings the connection with the Baltic, missing from IBI. The overlapping zone also offers interesting intercomparison possibilities between the two models. The authors could say a few words about the respective interests and the complementarity of NEMO-NORDIC and NEMO-IBI and if possible make a reference to the validation study of NEMO IBI by Maraldi et al, 2013.

Yes. The idea is that to have good North Sea baroclinic dynamics, you need a good Baltic Sea freshwater outflow, which most models don't have. And to get good Baltic Sea dynamics you need the North Sea variability. Not to mention the most interesting area of Nemo-Nordic which is Kattegat/Skaggerak. We have mentioned this aspect and added the relevant citation.

line 108: NEMO's recent advances on the sigma coordinate could be cited here. I think for example to the paper of Shapiro 2013 (or other if the authors see a reference more relevant)

Indeed, we have added this reference.

line 112 "adopted" "adapted"

Yes, thanks.

L132: the word "decouples" seems clumsy to the extent that there is in fact a coupling between baroclinic and barotropic modes.

Indeed, we'll correct this.

L135: the term "degree of conservation" implies that the conservation properties of tracers may not be strictly respected, which seems a priori not very compatible with the study of climate. If the model really respects the conservation properties of tracers the authors should say it more clearly (or can refrain from commenting on a fairly basic property).

Quite so. I think we shall refrain from commenting on this aspect.

L140: Is the roughness given by a length of roughness? Is it a constant or a mapped parameter? Since this parameter seems important it would be interesting to give its value, or its order of magnitude if it is not a constant.

In operational mode (1nm resolution), the roughness is mapped. It is constant for the longer term simulations. We will add a better description.

L145-150: without doubting the proper functioning of the BBL, is it still not unsatisfactory to have to completely remove the advective component of this parameterization? Independently of the numerical considerations that one has well understood, is it not less realistic from the point of view of physics? Would that not finally plead for the use of the sigma coordinate referred to online 107?

Actually the transition times of Baltic flows are several months, so a small advection speed might help. We need to investigate this matter further. Sigma coordinates would help for the overflows but would create many other problems, like destroying completely the Baltic Sea halocline. The best would be hybrid coordinates like in Getm. We'll add more lines about this.

lines 150-160 What is "tuning" mentioned by the authors is not very clear. Do the authors refer to the value of the Galperin coefficient? In this respect, the reference

to Galperin's paper is too vague. One could for example think that one refers to the functions of stability of Galperin? Is that the case? According to Reffray et al 2015 the choice of Canuto seems the most judicious... Or do you refer to the limitation of the mixing length (eq22 of Galperin)? What exactly is this coef mentioned above? The problem raised by the authors also refers to the choice of the thresholds of minimum values for the TKE of the closure scheme and their possible regionalization. Can the authors say a few words about the values used by NEMO?

We refer to the mixing length limitation, itself tuned with the Galperin's coefficient. We have made this more precise.

lines 165-175 The authors mention the drawbacks of the calculation of the horizontal mixing in z coordinate, which, even taking into account the NEMO rotation tensor, tends to introduce a significant diapycnal mixing. The authors correct this defect by means of a spatial adaptation of the coefficients of viscosity / diffusivity which seems a little artificial but which has the merit of working. There is certainly room for discussion of possible future improvement prospects for NEMO-NORDIC. Insofar as the saline intrusions evoked by the authors would follow the bottom, one can for example wonder if the sigma coordinate would not be better adapted. The work of Shapiro 2013 on the different forms that the sigma coordinates can take in NEMO could be a source of inspiration for a possible evolution of NEMO-NORDIC in this direction.

As said before, sigma would be better for the overflows, but would have dire consequences for the frontal structures and the permanent stratification. The best would be hybrid coordinates, but this is for the future. Meanwhile, we believe there are possible improvements to do with our viscosity/diffusivity coefficient.

line 183: the tide is apparently introduced as a boundary condition only. Does this mean that the internal generating forces in the numerical domain (astronomical potential and loading self attraction present in the NEMO version used by Kodaira et al, 2016) are not used here? If yes, why? Are they negligible in comparison with the influence of

boundary conditions?

Yes, we have neglected the tidal potential. NOAA on its website says US great lakes have a maximum spring tide of 5cm so it could be worth trying. So far our tidal signal is actually too strong, but we will try this sensitivity experiment in the future.

lines 184 The open boundary conditions seem to have a fairly high level of elaboration with respect to the barotropic processes (tide, storm surge). On the other hand, I am surprised, given the possible operational purpose, and also given the Copernicus context in which the NEMO-NORDIC model seems to be developed, by the great simplicity of the boundary conditions for the general 3D circulation. Only temperature and salinity seem to be concerned (nothing specific is said about SSH and currents, apparently). In addition, T and S would be climatological. This simplicity can be understood in the context of a climate projection, but for operational applications it is expected that the Coperninus operational system will serve to provide boundary conditions for regional models such as NEMO-NORDIC. I may have misunderstood the text which in this case should be a little clarified. Note also that the IBI operating system seems to have the capacity to forecast storm surges according to Maraldi et al 2013. Can the authors discuss a little more about their choices?

Indeed, thanks for this remark, this is a mistake of ours when drafting the manuscript. In operational mode, having proper OBCs has been a subject of great concern, especially for the barotropic mode. We will add more details.

line 185. The authors apparently use the TPXO tide atlas of Egbert et al 1994. In Maraldi 2013, the accuracy of altas FES is widely commented and finally used as a reference to validate the quality of the tide simulation obtained with NEMO. The present study could have been an opportunity to make a comparison between the different tidal atlases usable as boundary conditions. Which produces the best result? etc etc. . . That would be useful I think. This is a minor remark but if the authors deem it appropriate they might mention this fact as a possible prospect in future work.

Absolutely, it is just a question of time: one of the co-authors of this article has suggested we use FES. We just did not have the time to try, but will mention it.

line 195-205 The introduction and the abstract of the article suggest that NEMO-NORDIC is used for both climate studies and short-term operational forecasting. The description of the atmospheric forcing seems to correspond to the first point only. What is the authors' strategy for short-term operational forecasting? Does the hourly forecast of the sea level for example impose particular constraints with regard to the frequency of the atmospheric forcing? We can also think that the precise prediction of the sea level requires taking into account the effect of the waves. Preliminary developments have been made in NEMO on this subject (see, for example, NEMO and WW3 coupling by Clementi et al, 2017 for a better representation of the drag coefficient). What is the authors' strategy for this question?

Indeed, we had forgot to mention the forcing used in operational mode. This is now corrected. There has not been a thorough investigation on the influence of the frequency of the atmospheric forcing so far, the goal has been to find the best accessible data to obtain results which fit the quality requirements of the model in forecast mode, through a benchmark. About the coupling with wind waves, this work is ongoing at the Finnish Meteorological Institute.

lines 208-215 About taking into account a constant concentration of chlorophyll to improve the essential point of the penetration of light. Would there be an interest (perspective) in using Copernicus' global predictions of chlorophyll?

It would be interesting for future studies but we have not pushed our investigation so far. Basically we provided a value which is different than the NEMO default value to take into account the Baltic & North Sea turbidity.

line 243: I am surprised when the authors say that the correlation is mostly close to 0.99. I would have rather said 0.95. This difference of appreciation is probably subjective and attributable to the lack of readability of Figures 2-3-4. In fact it seems to

me that Taylor diagrams are not very suitable here. Figures 2-3-4 indeed occupy a lot of space for little information (only two points, a yellow, a blue) with a very low level of readability since each individual figure per tide station is finally tiny. We therefore lose a lot of time trying to see what are the RMS values, Standard deviation, correlation, when a simple table would immediately give this information, and allow a quick comparison with other authors (see for example Table 1 in Maraldi 2013). It seems to me that Taylor diagrams are appropriate when a single reference is compared to a scatter plot. For example, in Toublanc et al (2018) Figure 7, it is immediately understood that the simulations corresponding to green and blue point clouds are better than the simulation

We have converted this data into arrays.

line 247: The issue of the horizontal resolution is appropriately addressed in the Strait of Denmark. Some passages are indeed so narrow that a resolution of 2nm (almost 4km) seems clearly insufficient. For example the passage between Elsinore and Helsingborg barely fits a mesh. However, in NEMO, there is possibility for local increase in horizontal resolution, either by using an AGRIF nesting (Waldmann et al, 2016), or by using the NEMO curvilinear grid (Madec Imbard, 1996). To what extent can either of these two possibilities constitute a NEMO-NORDIC development perspective?

AGRIF is the way to go, but that is for the future.... Actually I am right now just back from the Nemo User's Meeting and it seems finally that Agrif works with the non-linear free surface in Nemo (key_vvl), which was not the case so far. The non linear free surface being an essential feature of Nemo Nordic, we did not spend too much time investigating Agrif before being should it would work.

The thresholding of the bathy (note that Maraldi 2013 also uses a threshold and discusses its consequences) also seems problematic: is not it a handicap for the forecast of surges? What is the technical reason that prevents lower bathymetry? Could the sigma coordinate overcome this problem?

We are not sure to understand, we do not limit the value of the bathymetry: the only

treatment of the original database we perform is to interpolate it on the Nemo-Nordic grid.

Table 1: Northern boundary, the "Inflow Observations" bounds are given in descending order. Is it correct?

It was not very clear indeed, and there was a bug in one of the values. We changed that. The values given are a range, and are now in ascending order.

Line 328. My next question is motivated by the authors' commentary on the Galperin coefficient and the fact that NEMO has two types of turbulent closure (TKE, k-epsilon, Reffray 2015). In the manner of Reffray et al 2005, have the authors made a sensitivity test of the SST bias to the turbulent closure scheme (TKE or K-epsilon)?

We have not made such an experiment with Nemo-Nordic, but we tried different values of the Galperin coefficient.

lines 353-355: In the description of the model it would therefore be interesting to say a few words about the state equation used.

Indeed, we have added a line when providing the runoff description.

line 370. It seems to me that the mechanisms responsible for the Major Baltic Inflows of 1993 and 2003 could be explained in more detail if possible (would there be no more things to say outside of the sea level? ?). Do these mechanisms come from open boundary conditions, or are they generated inside the modeling domain etc etc ...?

This is a very nice remark, but still a subject of intense debate among researchers. We have added a line

line 393. Specify the number of the Figure

Corrected.

line 415: the authors evoke the possibility of a validation of their boundary conditions.

But what exactly are the boundary conditions for T and S? How are the T and S fields constructed that force the open-border model? Climatology, ORCA25? Same remark for baroclinic currents.

The sentence is not very clear indeed, we changed it. The data source for T&S at the open boundary is a climatology for our long term hindcast, but can be anything else depending on the simulation. We do not use baroclinic currents data for the open boundary conditions, but a simple radiation condition.

Figure 14: Would a single color palette (as in Figure 15) be preferable?

We have tried to adapt the color scale to the salinity range, the idea was to visualize as best as possible the biases of the model when comparing with observations.

line 481: it seems to me that the acronym BSH is not defined.

Yes, this is corrected.

Please also note the supplement to this comment:
https://www.geosci-model-dev-discuss.net/gmd-2018-2/gmd-2018-2-AC1-supplement.pdf

---

## Author Comment (AC2) · 29 Oct 2018

**Revised Version of the Manuscript in File Attached**

First, we want to thank you for your work on our manuscript, we have tried to correct the many typos and hope this latest version will meet the quality standard expected for publication in GMD.

I think within the introduction (L85) or the Model set up needs mentioning of the two differing resolution models that are referred to later in the paper. It is not immediately obvious which version of Nemo-Nordic is being assessed at any one time, especially

as both are later compared against each other. I think it would help the reader if there was some way to make this clearer, e.g. Nemo-Nordic 1nm/2nm etc. or some other similar labelling strategy early in the paper and a description of these. In the model description there is only a description of the 2nm version. Perhaps restricting nautical miles to metric equivalents will be more in line with GMD.

Yes, indeed. We have now written clearly in the introduction that there are two configurations of different resolutions. Each time nautical miles units are used then the equivalent resolution in meters is now also written.

L105. With regards to the 2nm grid description, it might be useful to state if the grid is rotated, otherwise it would be hard to see how the stated grid resolution would be retained at a relatively high latitude.

We have made this more precise. The grid is un-rotated, it is a simple geographical grid.

L113-L116 The stated vertical resolution is surprisingly coarse in a regional model. I appreciate there is a need to focus resolution with regards to the overflows but 3 m surface resolution seems quite low. I refer the authors to Stewart et al. with regards to what would be an optimal vertical resolution for a z-level model in a global context. K.D. Stewart, A.McC. Hogg, S.M. Griffies, A.P. Heerdegen, M.L. Ward, P. Spence, M.H. England,Vertical resolution of baroclinic modes in global ocean models,Ocean Modelling,Volume 113, 2017,Pages 50-65,https://doi.org/10.1016/j.ocemod.2017.03.012. Towards the end of the paper there is an analysis compared to an SST product. The bias is surprisingly large and cold given the warm bias in the atmospheric forcing, could the surface resolution play a part? What is defined as SST in this context ?

The vertical resolution is a compromise, which allows to have an acceptable resolution close to the surface but also closer to the bottom in order to resolve as best as possible the Baltic overflows. The overflows are driven by barotropic processes, and the halocline/thermocline is usually located far below the level the first grid cells, so we do not

believe this kind of bias is linked with the resolution. However there is a clear problem linked with the Galperin parameterization, which we believe needs to be applied only to haline stratification, we are working with this issue.

L124 The issue of model resolution and the Danish straits is correctly brought to the attention of the reader and the method by which the barotropic flux can be maintained by retaining the same cross-sectional area. However, this must be problematic with regards to the baroclinic part of the flow. Particularly so as one of the main motivations of having the interconnect is to model bottom saline intrusions form the North Sea that enter the Baltic. Perhaps there is justification here for some more comment on the effects on the baroclinic flows by attempting to retain the barotropic flux.

Indeed, we have added a few lines about this specific issue.

L136 It is mentioned that 'tuning' is done with regards to optimizing model SSH. It is not clear what the optimization is, perhaps this could be elaborated as it could potentially save others time in the future or suggest useful strategies. I wonder could the authors supply a graphic/map in the supplementary material with regards to the 2d varying bottom friction "following the barotropic Kelvin wave", what is the physical grounds for this?

We can provide the input file that is used for this tuning, but the entire model is available on demand actually, including this file. The underlying idea is that barotropic waves entering the North Sea have a high energy that they lose while propagating cyclonically along the coastline. The bottom roughness tuning is done to fit the loss of energy.

L174-L175 The use of variable diffusivity and viscosity appears to be an interest- ing pragmatic engineering solution to the model difficulties concerning mixing and the dense water overflows. That is an interesting solution and appropriate for short time scale like a forecast model, but I wonder if it is appropriate for climate scales? That is this strategy assumes a-priori what the structure of the water column is, but on climate time scale that could change but the model may in effect be imposing it as it is, could

the authors comment on this. It seems that as the authors note, a hybrid z* with sigma at the bottom is a much better vertical framework for the problem at hand. Could the authors comment on why such a huge viscosity is required at the boundary region? This is likely to cause severe issues for any coupled biogeochemistry model here. I suggest it is worth investigating what is happening to vertical velocity and tke here.

Actually this idea is made more for climate time scales (i.e.: several decades) which affect the Baltic Sea salinity. As long as the Baltic Sea is a semi enclosed basin with low turbulence this would work. We have added a few lines in the manuscript. Sigma coordinates would be better at the bottom indeed but would have dire consequences at the level of the halocline, the best would be hybrid coordinates but long term simulations would be very costly. z* coordinates are not perfect but a good compromise. High viscosity at the OBCs does not affect areas of concern for biogeochemistry so far, which we always manage as for the physics to be far away from these OBCs.

L210 There are a number of chl products that are available. Do the authors consider using say even just a monthly climatology rather than a uniform value domain wise.

Indeed, but we have not investigated this issue yet. So far our only concern with chlorophyll data is to get a light penetration that corresponds to reality, especially in the Baltic Sea.

Fig 2,3,4 I think the use of Taylor plots here is not appropriate, as there are only two data points. It could save a lot of space to reduce the Taylor plots to numerical tables. Taylor plots are beneficial when analysing a large 'cloud' of data. In a model sensitivity, they are useful if tuning say one parameter a number of times. In this case, there are just 2 model resolutions, it might be more appropriate if there are several model resolutions to intercompare. Enabling the modeller to visualise if there are say competing trends between rms, correlation and standard deviation. However, with just two data points there can be no trend to discern. In conclusion, a table might be quicker to interpret and save considerable space I do not think the Taylor plots here bring

any advantage. With regard to the tides in the North Sea, the inclusion of a Co-tidal amp/phase plot of say M2 could be useful to give a quick look at how overall the model is doing in space with regards to tides.

The other reviewer made the same remark, we replaced the Taylor diagrams by arrays.

Fig 5 The climatological currents from North to South along the boundary in the English channel are very odd and suggest some problem in the bdy implementation here. Could the problem be related to using clim. TS in a highly tidal area? Ignoring the general cold bias in Fig 17, there is still an obvious bdy issue both in the north and in the south in JJA, again perhaps relates to the bdys provided. Too much vertical mixing??

Basically what the figure shows are the mean currents of a place where there is a huge variability compared with the mean value. Along the Western side of the English channel opening towards the Atlantic ocean, the mean currents are not entirely along the main direction of the channel as it is the case in many places in the channel itself. The flow being almost entirely barotropic in such a region, if there was such a mistake in the model it would show immediately in the sea level.

L256 It is noted that the north sea underestimate lower frequencies but these are unbiased in the Baltic. (due to amplification?) Is there a case to be made that the model Is overdoing amplification of waves that are initially underestimated in the North Sea? If so could that have other adverse effects?

This is a very interesting question, which is difficult to answer here because it is a research topic in itself. Basically our understanding is that some low frequency waves coming from the Atlantic ocean enter the domain and are not included in our set of open boundary conditions. The effects on the Baltic Sea are difficult to estimate, so far it seems we are able to reproduce all the major Baltic inflows.

L 425 the authors show a large freshwater bias at the Frisian front location. May I also suggest that the riverine input from HYPE could be a possible issue here, Have the

authors made an assessment of the HYPE model along this coastline? The accuracy is assessed for the Baltic Basin but not for the North Sea?

Actually we have computed the mean value of of the E-Hype flow for the North Sea and it looked quite correct although there was a positive bias indeed. We have tried a lot of different forcing datasets for hydrology, including trying to correct biases in HYPE, all lead the same salinity bias (too fresh) . We believe a bias of circulation and/or mixing is the cause of this issue.

3 Minor Technical points/errata These are likely but a subsection of minor points that need further editing.

Thanks for this re-reading work, which helps a lot to correct the manuscript. We have taken one by one all the points and corrected them.

Please also note the supplement to this comment:
https://www.geosci-model-dev-discuss.net/gmd-2018-2/gmd-2018-2-AC2-supplement.pdf
* * *

---

## Author Response (AR2)

Robinson Hordoir, PhD
Institute of Marine Research
P.O. Box 1870 Nordnes, 5817 Bergen, Norway
Mobile: +47 906 27 189
robinson.hordoir@hi.no

Prof. James R. Madisson
Editor for Geosciences Model Development
University of Edinburgh
United Kingdom

Bergen, November 21st 2018

Dear Prof. Madisson,

Please find attached a second revised version of the manuscript entitled "Nemo-Nordic: A NEMO based ocean model for Baltic & North Seas, research and operational applications".
After your request for minor revision, we have taken into account all your comments for which we provide a detailed answer.
Thanks a lot for all the work done on our manuscript.

Best regards,

Robinson Hordoir

**Response to comments**

*\* Figures 1 and 2 seem to contain copyrighted material. Note that you will need to have permission from the copyright holders for these to be released under the appropriate CC BY license
(see https://publications.copernicus.org/for_authors/licence_and_copyright.html), and in order for these figures to be used.*

I now have got emails mentioning explicitly I am given authorization to reproduce this material.

*\* Could you clarify the license of Nemo-Nordic, and is it possible to add a single copy of the code to a ZENODO repository ?*

Yes, this is now clarified. Nemo-Nordic is released under the Cecill license already used for NEMO. We also have uploaded the code into a Zenodo repository which link is now provided in the article.

*\* Please add an "author contribution" section, as described in the guidelines for authors
([https://www.geoscientific-model development.net/for_authors/manuscript_preparation.html](https://www.geoscientific-model%20development.net/for_authors/manuscript_preparation.html)).*

Done.

*\* Please provide more descriptive captions for tables 1, 2, and 3 -- I found it difficult initially to identify the quantities listed. The captions and main text refer to "diagrams"/"figures"/"arrays" (lines 277, 281, 282, 291) instead of "tables". There are inconsistencies between the values in the main text (0.94 and 0.6) and those listed in the tables (0.93 for 1nm, and 0.68 for 2nm).*

We fixed that.

*\* Reviewer #1 requested a figure for tidal amplitude and phase, but this comment does not seem to appear and be responded to in your response. A similar comment was made by reviewer #2.*

It was suggested indeed. We thought it was not compulsory and found it quite an important amount of work, so we neglected this suggestion. I think the North Sea statistics proved the point.

*\* I believe reviewer #1 is requesting more precise details regarding the Galperin parameterisation (e.g. a definition of the coefficient).*

Now we have added the explicit equation, with detail of every term including the coefficient.

*\* I could not locate the "line" added in response to the line 370 (old version line number), "Major Baltic Inflows" comment from referee #1.*

There was more clarity added somewhere else in the text, but now we have also added more explanation in this precise place in the manuscript to ease the transition to comparison of sea level during Major Baltic Inflows.

*\* The line 415 (old version line number), "validation of boundary conditions" sentence referred to by reviewer #1 does not seem to be changed as suggested by your response.*

This is corrected.

*\* The sentence starting on line 81 could perhaps be clarified.*

The following sentence: "Nemo-Nordic is declined into two resolutions of 2 nautical miles (3704 m) and 1 nautical mile (1852 m)." ?

*\* Can you comment on the large boundary viscosity, referred to by reviewer #2?*

Yes, we added a line " This value is huge but prevented the model to ever crash at the open boundary conditions. Several values were tried and it proved not have little effect on the sea level variability inside the model domain."

*\* Some cases of "Nemo" (rather than "NEMO") remain.*

Now I ran grep commands, either we write about the ocean engine and then it is "NEMO" or we write about our configuration and then it is "Nemo-Nordic".

*\* Should the title contain a comma?*

I think so if we keep the title as it is, but we could also change the title to something like "Nemo-Nordic 1.0: A NEMO Based Ocean Model for Research and Operational Applications in the Baltic & North Sea area". I don't really know.

*Can you clarify the sentence starting on line 95? What does "this latest model" refer to?*

Corrected.

*Similarly on line 121, can you clarify what "this latest configuration" refers to?*

Corrected

*Figures 8, 9, 13, and 14 refer to "left column" when "right column" is intended. Figures 13 and 14 refer to "transparent circles" which do not appear in the figures.*

Corrected

*Both K and degree C are used in the text / figure 15 caption.*

Corrected

**Compare Results**

| Old File: | | New File: |
|---|---|---|
| **gmd-2018-2-AC1-supplement.pdf** | versus | **NEMO-Nordic_validation-2.pdf** |
| 31 pages (9,86 MB) | | 31 pages (9,90 MB) |
| 29/10/2018, 17:13:03 | | 21/11/2018, 17:50:40 |

**Total Changes**

**321**

Text only comparison

**Content**

59 Replacements

132 Insertions

130 Deletions

**Styling and Annotations**

0 Styling

0 Annotations

Go to First Change (page 1)

[revised manuscript text omitted]

---

## Author Response (AR3)

Robinson Hordoir, PhD
Institute of Marine Research
P.O. Box 1870 Nordnes, 5817 Bergen, Norway
Mobile: +47 906 27 189
robinson.hordoir@hi.no

Prof. James R. Madisson
Editor for Geosciences Model Development
University of Edinburgh
United Kingdom

Bergen, Dec. 10th 2018

Dear Prof. Madisson,

Please find attached the revised version of our Nemo-Nordic GMD article. We have taken into account your corrections, for which we provide an answer below.

Thanks to the editorial team, to the reviewers and to you, for helping us improve this manuscript.

Best regards,

Robinson Hordoir

**Response to comments**

*I am not satisfied that the necessary permissions for figure reproduction have been satisfied. For figure 1, as this from a Springer publication, please see: https://www.springer.com/gb/rights-permissions/obtaining-permissions/882. For figure 2, the CC BY-NC is not appropriately compatible with the CC BY license of the journal. Please seek more definitive permission, or else consider the necessity of including these figures.*

Getting the proper permission from the authors is difficult. We simply removed the figures.

*Please add the doi for the relevant Zenodo archive to the "Code and data availability" section.*

This is now added.

*I think either "K~m^{-1}" or "\circ C m^{-1}$ are preferable in the figure 15 caption. My earlier comment was to note that both K and degrees C are used in the text.*

We have now written "C$^{\circ}$ m$^{-1}$".

**Compare Results**

| Old File: | | New File: |
|---|---|---|
| **gmd-2018-2-manuscript-version7.pdf** | versus | **NEMO-Nordic_validation-3.pdf** |
| **31 pages (9,90 MB)** | | **30 pages (9,62 MB)** |
| 21/11/2018, 17:50:40 | | 10/12/2018, 09:55:36 |

**Total Changes**

**250**

**Content**

37    Replacements

84    Insertions

99    Deletions

**Styling and Annotations**

21    Styling

9    Annotations

Go to First Change (page 1)

[revised manuscript text omitted]